# A General Single-Cell Analysis Framework via Conditional Diffusion Generative Models

## Abstract

The fast-growing single-cell analysis community extends the horizon of quantitative analysis to numerous computational tasks. While the tasks hold vastly different targets from each other, existing works typically design specific model frameworks according to the downstream objectives. In this work, we propose a general single-cell analysis framework by unifying common computational tasks as posterior modeling problems. In light of conditional diffusion generative models, we introduce *scDiff* through the proposed framework and study different conditioning strategies. With data-specific conditions, *scDiff* achieves competitive performance against state-of-the-art in various benchmarking tasks. In addition, we illustrate the flexibility of *scDiff* by incorporating prior information through large language models and graph neural networks. Additional few-shot and zero-shot experiments prove the effectiveness of the prior conditioner on *scDiff*.

## 1 Introduction

Recent advances in single-cell technology enable extensive computational tasks for quantitative understanding of underlying biological principles (Wen et al., 2022; Elmentaite et al., 2022; Heath et al., 2016). Some typical examples of these tasks include cell-level classification (Ma & Pellegrini, 2020; Xu et al., 2021), missing value imputation (Eraslan et al., 2019; Huang et al., 2018) and generalization to novel conditions (Hetzel et al., 2022; Roohani et al., 2023; Lotfollahi et al., 2019). Existing works often design distinct frameworks for different tasks according to their objectives. For example, cell type annotation algorithms (e.g., ACTINN (Ma & Pellegrini, 2020)), typically model the class attributes with multi-class cross-entropy loss; single-cell imputation methods (e.g., DCA (Eraslan et al., 2019)) aim to recover the true counts from dropouts by specifying some prior distribution; and perturbation prediction frameworks (e.g., GEARS (Roohani et al., 2023)) explicitly model the change in expression between perturbed cells and control cells.

In this work, we provide a new perspective by formulating common single-cell tasks through the lens of distribution modeling. While the tasks are derived from diverse biological perspectives, we articulate that their objectives can be described as posterior distribution modeling problems under task-specific conditions, as detailed in Section 2. For example, cell type annotation can be considered as classifying each cell to the cell type that maximizes the conditional likelihood of its expression (Li et al., 2023); and imputation can be treated as drawing samples from the learned posterior given the partially observed expression data. This perspective brings new opportunities to single-cell analysis. It allows a general posterior modeling framework that enables us to handle multiple single-cell analysis tasks with a single unified objective. In the meantime, many conditional generative models can be plugged into the framework. However, it also introduces new challenges. Within the framework mentioned above, the choice of the generative model plays a crucial role in how well they can conduct downstream tasks (Dhariwal & Nichol, 2021). An expected model would bring out accurate distribution estimation and high-quality sample generation. Meanwhile, better conditioning strategies are desired to trade off the influence across different conditions.

To address the challenges mentioned above, we delve into diffusion generative models (DGMs). DGMs have shown great successes in generation tasks since the introduction of denoising diffusion probabilistic models (Ho et al., 2020), which are further extended to conditional scenarios through classifier guidance (Dhariwal & Nichol, 2021) and classifier-free guidance (Ho & Salimans, 2021). Various types of conditions have been applied to guide the DGMs, like image (Poole et al., 2022),

text (Rombach et al., 2022; Kim et al., 2022b), and audio (Ruan et al., 2023; Leng et al., 2022). Compared to class guidance, those approaches paved the way to guide DGMs with prior knowledge. This flexibility allows us to construct the probabilistic modeling process through conditional DGMs in single-cell analysis with internal and external information. We demonstrate that internal-guided DGMs can match state-of-the-art performance in standard settings while prior knowledge enables better transferability under zero-shot and few-shot settings. We summarize our main contributions as follows:

- We present a general single-cell analysis framework by formulating various tasks as posterior modeling problems. Through this framework, we introduce *scDiff* with a conditional DGM.
- We study the conditioning strategies of *scDiff*. With cell-label conditioning, *scDiff* achieves competitive performance with state-of-the-art models in various benchmarking tasks.
- We incorporate prior knowledge with large language models (LLMs) and graph neural networks. Experimentally, *scDiff* shows outstanding few-shot and zero-shot results.

## 2 A GENERAL POSTERIOR MODELING FRAMEWORK

In this section, we detail our proposed framework. Section 2.1 aims to formulate the various single-cell tasks as posterior modeling questions. Section 2.2 introduces the background of conditional DGMs and the structure of the *scDiff*.

### 2.1 SINGLE-CELL TASKS AS POSTERIOR MODELING

Single-cell analysis is a vast topic that involves a large number of computational and biological tasks. While existing works provide novel and effective solutions for individual tasks, few can address multiple challenges. A natural reason is that the tasks are quantifying distinct perspectives of the underlying mechanisms. Hereafter, we will argue that many common tasks in single-cell analyses amount to quantifying the cell identities given the biological context. In other words, we are actually estimating the posterior distribution of the cells' expression given specific conditions.

Formally, we denote the expression of all cells as $\mathbf{X} \in \mathbb{R}^{n \times m}$, where $n$ is the number of cells and $m$ stands for the number of genes. We categorize the common tasks into three classes: cell labeling, expression completion and knowledge transfer. The notations for task-specific conditions are detailed subsequently.

**Cell labeling.** One of the most critical single-cell tasks is to label cells by their expression. Specifically, let us denote the labels as $\mathbf{C}_{\text{label}} \in \mathbb{R}^n$, which can either be discrete (e.g., cell type) or continuous (e.g., spatial cell type ratio). For the cell labeling task, we estimate the posterior of the labels given the expression $\mathbf{X}$. Inspired by Li et al. (2023), we formulate the posterior with the Bayesian theorem

$$p\left(\mathbf{C}_{\text{label}} \mid \mathbf{X}\right) = \frac{p\left(\mathbf{C}_{\text{label}}\right) p\left(\mathbf{X} \mid \mathbf{C}_{\text{label}}\right)}{\mathbb{E}_{\mathbf{C}_{\text{label}} \sim p_{\mathbf{C}_{\text{label}}}}\left[p\left(\mathbf{X} \mid \mathbf{C}_{\text{label}}\right)\right]} = \frac{p\left(\mathbf{X} \mid \mathbf{C}_{\text{label}}\right)}{\int p\left(\mathbf{X} \mid C_{\text{label}}\right) dC_{\text{label}}}, \tag{1}$$

where we assume a uniform prior on the support of $\mathbf{C}_{\text{label}}$. Consequently, the problem shifts from estimating label posterior to approximating expression posterior.

One representative task of cell labeling is cell type annotation, where the labels become the functional types of the cells. The integration in the denominator of equation 1 then reduces to summation on the classes. From the data perspective, the ground truth annotation is typically given by experts' curations. Most existing works directly model the posterior of the labels with multi-class cross-entropy loss. Another typical task is cell trajectory inference, where the labels become the developmental trajectories of biological progression through processes such as cell differentiation (Qiu et al., 2022). Similar ideas can be extended to spatial transcriptomics. For example, for cell type deconvolution, the labels become the ratios of cell types within each spot (Biancalani et al., 2021; Ma & Zhou, 2022). More details can be found in Appendix A

**Expression completion.** A crucial category of single-cell analysis tasks is expression completion. It includes both filling in missing values or predicting the whole expression. In some scenarios, the task may require external information from reference datasets. To account for the majority of

the settings, we denote the observed expression as $\mathbf{M} \odot \mathbf{X}$, where $\odot$ denotes matrix element-wise multiplication and $\mathbf{M} \in \{0, 1\}^{n \times m}$ is the element-wise indicator with ones be observed and zeros be missing. The task is defined to estimate the posterior $p((\mathbf{J}_{n,m} - \mathbf{M}) \odot \mathbf{X} \mid \mathbf{M} \odot \mathbf{X})$, where $\mathbf{J}_{n,m}$ is an all-ones matrix of dimensions $n \times m$. Equivalently, we write the objective as

$$p(\mathbf{X} \mid \mathbf{M} \odot \mathbf{X})^1. \tag{2}$$

In missing value imputation, the partially observed expression is produced by manually dropping out none-zero counts (Eraslan et al., 2019; Van Dijk et al., 2018). The dropout mechanism can be extended to molecular cross validation (Batson et al., 2019), where a continuous $\mathbf{M} \in [0, 1]^{n \times m}$ becomes the element-wise dropout rate specified by the partition of molecules. When some genes are completely unobserved, the objective shifts to missing gene imputation (Arisdakessian et al., 2019; Biancalani et al., 2021). Specifically, in equation 2, the indicator matrix $\mathbf{M} = [\mathbf{J}_{n,s}, \mathbf{O}_{n,m-s}]$ describes $s$ seen genes and $m - s$ unobserved genes, where $\mathbf{O}_{n,m-s}$ is an all-zeros matrix with dimensions $n \times (m - s)$. In the multiomics setting, the previous formulation can be extended to modality prediction tasks, where the objective is to predict the expression of the target modality from source modality (Yang et al., 2021; Wu et al., 2021). We detail the formulation in Appendix A.

**Knowledge transfer.** While the exact condition control in single-cell experiments is both time-consuming and resource-consuming, it is desired to have the computational method to transfer known results to unseen conditions (Roohani et al., 2023; Hetzel et al., 2022). Let $\mathbb{C}_s$ be the source condition set and $\mathbb{C}_t$ be the target condition set with $\mathbb{C}_s \cap \mathbb{C}_t = \emptyset$, we aim to estimate the expression under target conditions. External reference conditions may also be included to enhance the estimation with prior information. The knowledge transfer task can be formulated as:

$$\text{Estimate } p(\mathbf{X} \mid \mathbb{C}_t) \text{ given } p(\mathbf{X} \mid \mathbb{C}_s). \tag{3}$$

Common single-cell knowledge transfer tasks mainly focus on perturbation prediction, which includes predicting novel gene perturbation responses (Roohani et al., 2023), predicting novel drug perturbation responses (Hetzel et al., 2022) and cell type transfer of perturbation (Lotfollahi et al., 2019). Considering the first two tasks, prior knowledge is often needed to generalize to unseen perturbation types. For the remaining task, we focus on only one type of perturbation and aim to generalize the perturbation effect across cell types. Specifically, the training set contains both perturbed and control cells for the source cell types but only control cells for the target cell types, and the model is expected to approximate the perturbed state of the target cell types.

**A general objective.** The objectives of the aforementioned tasks are to model the posterior distribution of the cells' expression given task-specific conditions. Thus, a general objective can be formally stated as:

$$p(\mathbf{X} \mid \mathbf{C}), \text{ where } \mathbf{C} \in \mathbb{C} = \{\mathbf{C}_{\text{label}}\} \cup \{\mathbf{M} \odot \mathbf{X}\} \cup \mathbb{C}_s. \tag{4}$$

Without loss of generality, we write $\mathbf{x}$ as $\mathbf{x}_0$ on sample level. In the following section, we focus on objective $p(\mathbf{x}_0 \mid \mathbf{c})$ for one cell.

## 2.2 Preliminary of Conditional Diffusion Model

To model the posterior distribution, we delve into Diffusion generative models (DGMs). Following the notations in denoising diffusion probabilistic models (DDPM) (Ho et al., 2020), we denote $\mathbf{x}_{1:T}$ as the latent variables that have the same dimension as $\mathbf{x}_0$. Utilizing the Bayesian theorem, we model the posterior by a Markov chain:

$$p_\theta(\mathbf{x}_0 \mid \mathbf{c}) = \int_{\mathbf{x}_{1:T}} p_\theta(\mathbf{x}_{0:T} \mid \mathbf{c}) \, d\mathbf{x}_{1:T}, \text{where}$$

$$p_\theta(\mathbf{x}_{0:T} \mid \mathbf{c}) = p(\mathbf{x}_T) \prod_{t=1}^{T} p_\theta(\mathbf{x}_{t-1} \mid \mathbf{x}_t, \mathbf{c}). \tag{5}$$

The above process (i.e., *reverse process* in DDPM) learns to recover the original data from Gaussian white noise. Conversely, the *forward process* gradually corrupts data by adding Gaussian noise:

$$q(\mathbf{x}_{1:T} \mid \mathbf{x}_0) = \prod_{t=1}^{T} q(\mathbf{x}_t \mid \mathbf{x}_{t-1}), \quad q(\mathbf{x}_t \mid \mathbf{x}_{t-1}) = \mathcal{N}\left(\mathbf{x}_t; \sqrt{1 - \beta_t}\mathbf{x}_{t-1}, \beta_t \mathbf{I}\right), \tag{6}$$

---

[1]Empirically we approximate the posterior via conditioning. See Appendix D.2.

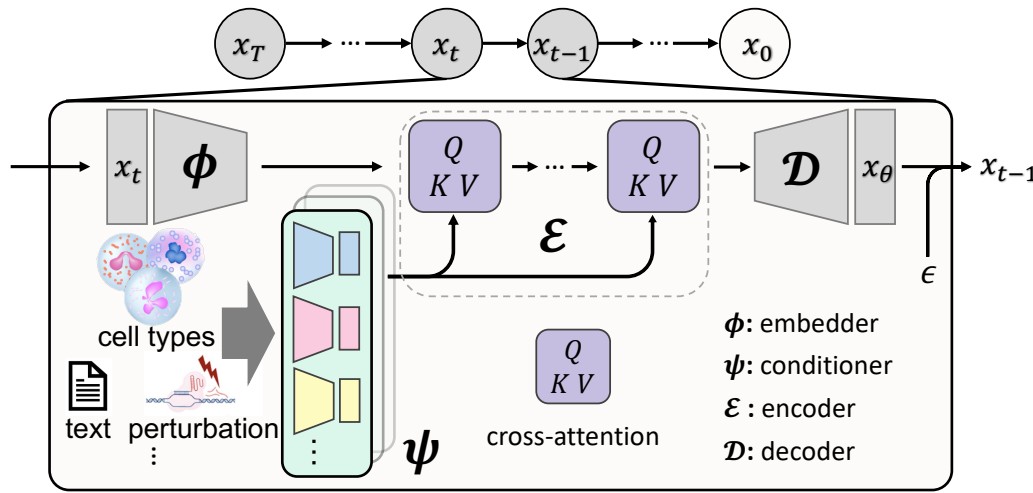

Figure 1: An Overview of *scDiff*.

where $\{\beta_t\}_{t=1}^T$ is the variance schedule. Next, we detail the parameterization of the *reverse process*. We empirically find that the widely-used predict-$\epsilon$ objective fails to recover the expression. Since single-cell data often shows extreme sparsity with more than $95\%$ of the entries as zeros, the corrupted input at time-step $t$, i.e., $\mathbf{x}_t$, will mostly be pure noise. Under predict-$\epsilon$ parameterization, the model would likely learn to reverse the noise schedule instead of modeling the data posterior. Therefore, we turn to predict-$\mathbf{x}_0$ parameterization. Denote $\alpha_t = 1 - \beta_t$ and $\bar{\alpha}_t = \prod_{s=1}^t \alpha_s$, we write the *reverse process* as:

$$p_\theta\left(\mathbf{x}_{t-1} \mid \mathbf{x}_t, \mathbf{c}\right) = \mathcal{N}\left(\mathbf{x}_{t-1}; \boldsymbol{\mu}_\theta\left(\mathbf{x}_t, \mathbf{c}, t\right), \sigma_t^2 \boldsymbol{I}\right), \text{where}$$
$$\boldsymbol{\mu}_\theta\left(\mathbf{x}_t, \mathbf{c}, t\right) = \sqrt{\bar{\alpha}_t}\mathbf{x}_\theta\left(\mathbf{x}_t, \mathbf{c}, t\right) + \sqrt{1 - \bar{\alpha}_t}\epsilon. \tag{7}$$

Due to the integral in equation 5, the data posterior is intractable. Alternatively, the parameters are optimized by minimizing the variational lower bound (ELBO):

$$\mathbb{E}\left[-\log p_\theta\left(\mathbf{x}_0 \mid \mathbf{c}\right)\right] \leq \mathbb{E}_q\left[-\log \frac{p_\theta\left(\mathbf{x}_{0:T} \mid \mathbf{c}\right)}{q\left(\mathbf{x}_{1:T} \mid \mathbf{x}_0\right)}\right]. \tag{8}$$

As detailed in Appendix B, we start from the ELBO and arrive at the simplified training objective:

$$\mathbb{E}_{t,\mathbf{x}_0,\epsilon}\left[\left\|\mathbf{x}_0 - \mathbf{x}_\theta\left(\sqrt{\bar{\alpha}_t}\mathbf{x}_0 + \sqrt{1 - \bar{\alpha}_t}\epsilon, \mathbf{c}, t\right)\right\|^2\right]. \tag{9}$$

### 2.3 *scDiff* MODEL ARCHITECTURE

We next introduce our *scDiff* model architecture, which is depicted in Fig 1. From a high level, *scDiff* aims to recover the clean single-cell gene expression $\mathbf{x}_0$ given the corrupted signal $\mathbf{x}_t$ with added Gaussian noise to time step $t$. The associated conditions of the cell are also fed into the model to provide conditional information. Specifically, *scDiff* follows a general encoder-decoder design and consists of four main components: (1) input expression *embedder* $\phi$; (2) various *conditioners*, $\psi_*$, where each converts a specific condition of the input cell into a sequence of dense numerical vectors; (3) a cross-attention *encoder*, $\mathcal{E}$, which combines the input embeddings with the corresponding conditioners and transforms them into the hidden representation of the input cell; and (4) a linear *decoder*, $\mathcal{D}$, that projects the hidden representation back to the gene expression space to recover the input cell's noise-free expression. We detail each component in the following.

**Embedder.** We use a linear mapping $\mathbf{W} \in \mathbb{R}^{m \times d}$ to project the noised gene expression $\mathbf{x}_t \in \mathbb{R}^m$ into $\mathbb{R}^d$ and further mix it with the sinusoidal time embedding (Appendix C.1) to inject diffusion time step information, following previous work (Ho et al., 2020).

$$\phi(\mathbf{x}_t, t) = \mathbf{x}_t\mathbf{W} + \text{TimeEmbed}(t). \tag{10}$$

**Conditioner.** The goal of each conditioner is to extract a set of $L$ numerical representations of an input condition $\mathbf{c}$, where $L$ is the number of unique conditions. Among these representations, each is used as the basis for the key and value embeddings in the cross-attention encoding step, as described

in the next section. Formally, each conditioner $\psi_f$ is a multilayer perceptron (MLP) that converts the raw representation extracted by $f$ into the final condition representation set[2].

$$\psi_f(\mathbf{c}) = \left\{\psi_f^{(i)}(\mathbf{c})\right\}_{i \in 1,\ldots,L} = \left\{\mathrm{MLP}_i\big(f(\mathbf{c})\big)\right\}_{i \in 1,\ldots,L}. \tag{11}$$

where $f \in \mathcal{F}$ is a function that maps an input condition into a $d$ dimensional vector and $\mathcal{F}$ is the set of mappings. The mapping here can be designed to suit the specific needs of different input types, including partially masked expression, class labels of cells and external prior information.

*Context.* A cell context is a randomly masked expression $\tilde{\mathbf{x}} = \mathbf{m} \odot \mathbf{x}$, where $\mathbf{m} \in \{0,1\}^m$ is the element-wise mask indicator. We process the context condition similarly as the input embedding using a linear projection, but without the time embedding: $f_{\mathrm{ctxt}}(\tilde{\mathbf{x}}) = \tilde{\mathbf{x}}\mathbf{W}_{\mathrm{ctxt}}$.

*Class.* We use learnable $d$ dimensional embeddings to represent each class, $f_{\mathrm{cls}}(\mathbf{c}) = \mathbf{h_c}$. The class attribute is an important information of the input sample, and can be used to guide the diffusion generation process (Ho & Salimans, 2021). In our case, class attributes can describe the cell type or the perturbation state of a given cell.

The two conditioners above utilize internal information obtained from the given training data. However, effectively incoporating prior information into the model is the key to more generalizable and transferable knowledge. We next describe two distinct approaches to incorporate prior knowledge as examples to illustrate the extendability of the conditioners.

*LLM.* Besides encoding the cell type attribute using class embeddings, we can alternatively leverage LLM to extract rich representations of different cell types using their textual descriptions. Specifically, the cell type definitions are first obtained from the cell ontology (Bard et al., 2005). We then feed these descriptions into the pre-trained BioLinkBERT (Yasunaga et al., 2022) model, and use the resulting class token embeddings as $f_{\mathrm{LLM}}(\mathbf{c})$.

*GEARS.* Roohani et al. (2023) proposed a novel approach to encode gene perturbation information using a graph neural network (GNN) on a gene similarity graph $\mathcal{G}$. This graph is constructed such that each edge represents the number of shared gene ontology terms (Ashburner et al., 2000) between each pair of genes, which reflects their functional similarities. The gene perturbation embeddings are then computed using simple graph convolution (SGC) (Wu et al., 2019), $f_{\mathrm{GEARS}}(\mathbf{c}) = \mathrm{SGC}(\mathbf{c}, \mathcal{G})$.

**Encoder.** Once the input embedding $\phi(\mathbf{x}_t, t)$ and the condition representations $\{\psi_f(\mathbf{c})\}_{f \in \mathcal{F}}$ are calculated, we combine them through multiple layers of cross-attention (Appendix C.3).

$$\mathbf{h}^{(l+1)} = \mathrm{CrossAttn}^{(l)}\big(\mathbf{h}^{(l)}, \big\{\psi_f^{(l)}(\mathbf{c})\big\}_{f \in \mathcal{F}}\big). \tag{12}$$

**Decoder.** Finally, the cell latent embedding is linearly projected back to the gene expression space to recover the clean expression signals $\mathbf{x}_0$. We follow Lopez et al. (2018) and mix in an additional learnable batch embedding with the latent embedding according to the batch label of the input cell. This approach can better disentangle the non-biological variations in the data.

$$\mathcal{D}(\mathbf{h}^{(L)}, \mathbf{c}_{\mathrm{batch}}) = \big(\mathbf{h}^{(L)} + \mathrm{BatchEmbed}(\mathbf{c}_{\mathrm{batch}})\big)\mathbf{W}_d. \tag{13}$$

Combining the above components, we summarize the full *scDiff* model in equation 14.

$$\mathbf{x}_\theta(\mathbf{x}_t, \mathbf{c}, t) = \mathcal{D}\Big(\mathcal{E}\big(\phi(\mathbf{x}_t, t), \{\psi_f(\mathbf{c})\}_{f \in \mathcal{F}}\big)\Big). \tag{14}$$

In Section 2.1 we establish a general objective in equation 4 by merging all the conditions into the prior. Consequently, in the context of DGMs, distinct tasks share a common training objective and necessitate varied inference directions. This facilitates a task-agnostic training process and diverse task-specific inference processes. More details are elaborated in Appendix D.

## 3 RELATED WORK

We introduce some existing works that study generative models in single-cell analysis. A large proportion of the single-cell generative models are variational autoencoders (VAEs). scVI (Lopez et al.,

---

[2]The only exception is the processing of the context embeddings, which we achieve by extracting the different hidden layers from a single MLP (Appendix C.2)

2018) led the trend of VAEs with a negative binomial prior to the raw expression. scVI achieved satisfactory integration results by explicitly incorporating library size and batch information in the model. Including those conditions helps to regress out the technological variance within the latent space. Many other existing works extended the design space of VAEs in single-cell analyes. scVAE (Grønbech et al., 2020) incorporated a Gaussian-mixture latent space to model the underlying clustering structure. scDHA (Tran et al., 2021) formed a hierarchical framework consisting of two VAEs. In a broader application scenario, scGen (Lotfollahi et al., 2019) utilized the VAE structure for out-of-distribution prediction, while scMM (Minoura et al., 2021) aimed at multiomics analysis with a mixture-of-experts VAE model. Meanwhile, several existing works applied generative adversarial networks (GANs) (Goodfellow et al., 2014) to single-cell analysis. cscGAN (Marouf et al., 2020) incorporated a conditional GAN for data augmentation. scIGANs (Xu et al., 2020) adapted GAN for single-cell imputation via generation. Despite existing works applying generative models in single-cell analysis, they are typically designed for one or a few tasks. This fundamentally limits their extendability to broader classes of problems.

## 4 EXPERIMENT

In this section, we conduct experiments to validate the effectiveness of *scDiff*. Statistics of datasets (e.g., the number of genes, cells, and cell types) and preprocessing pipeline are summarized in Appendix E.2. Through the experiments, we aim to answer the following research questions:

- **RQ1:** How does *scDiff* perform against the state-of-the-art with internal data-specific conditions?
- **RQ2:** Can *scDiff* extend to other application scenarios with external prior knowledge?

### 4.1 PERFORMANCE OF INTERNAL-CONDITIONED *scDiff*

To answer the first question, we choose one representative task from each of the three categories in Section 2.1, i.e., cell type annotation, missing value imputation, and perturbation prediction for novel cell type. It is worth noting that in this section, we implement *scDiff* with the same structure across three representative tasks. More implementation details can be found in Appendix E.1.

#### 4.1.1 CELL TYPE ANNOTATION

**Experimental settings.** Cell type annotation is one of the fundamental tasks in single-cell analysis. We collect 6 benchmark datasets: PBMC12K (Zheng et al., 2017; Lopez et al., 2018), Pancreas (Luecken et al., 2022), HLCA (Sikkema et al., 2023), Immune (Domínguez Conde et al., 2022), Brain (Seeker et al., 2023) and Liver (MacParland et al., 2018). We randomly hold out $10\%$ of the cells for each dataset as the test set and train all the models on the remaining cells. For *scDiff*, we annotate the cells by evaluating the mean square error between input expression and model posterior in a classifier-free approach (Li et al., 2023). We elaborate on the details in Appendix D.1. The classification results are quantified by the macro multi-class accuracy score and F1 score.

**Baselines.** We evaluate the performance of *scDiff* against the representative cell type annotation methods. The baselines are listed as follows. CellTypist (Domínguez Conde et al., 2022) is an automated tool for cell annotation based on logistic regression. SingleCellNet (Tan & Cahan, 2019) utilizes random forest along with top pair transformation. ACTINN (Ma & Pellegrini, 2020) is a three-layer MLP classifier. scANVI (Xu et al., 2021) is a variational autoencoder with auxiliary classifiers. All the baselines are evaluated based on their default settings provided by the authors.

**Experimental results.** Table 1 illustrates the cell type annotation results, where we report the macro accuracy scores with mean and standard deviation across five runs. Note that the colors in the result table in this section refer to the performance rank within one dataset, which is depicted as **first place**, **second place**, and **third place**. We highlight that *scDiff* achieves top performance in four out of six datasets without explicitly training a classifier, which extends the results in (Li et al., 2023). This observation offers a solid support for the proposed probabilistic modeling framework, suggesting that well-established generative models can even outperform discriminative models in the single-cell context. We include the macro F1 score results in Appendix E.3.

#### 4.1.2 MISSING VALUE IMPUTATION

**Experimental settings.** Missing value imputation aims to recover the true expression levels from the dropout events in sequencing (Hou et al., 2020). We choose three datasets, i.e., Jurkat, 293T,

Table 1: Macro ACC of cell type annotation

|  | PBMC12K | Pancreas | HLCA | Immune | Brain | Liver |
|---|---|---|---|---|---|---|
| Chance level | 0.4167 | 0.3353 | 0.1445 | 0.1125 | 0.4081 | 0.3064 |
| SingleCellNet | $0.8446 \pm 0.0064$ | $0.6436 \pm 0.0006$ | $0.8113 \pm 0.0046$ | $0.7752 \pm 0.0009$ | $0.8768 \pm 0.0033$ | $0.8723 \pm 0.0023$ |
| ACTINN | $0.6141 \pm 0.0709$ | $0.5277 \pm 0.0926$ | $0.2176 \pm 0.0440$ | $0.2361 \pm 0.0300$ | $0.6950 \pm 0.0624$ | $0.6138 \pm 0.0349$ |
| scANVI | $0.9297 \pm 0.0148$ | $0.9632 \pm 0.0083$ | $0.7081 \pm 0.0183$ | $0.8505 \pm 0.0133$ | $0.9325 \pm 0.0010$ | $0.9084 \pm 0.0144$ |
| CellTypist | $0.8828 \pm 0.0055$ | $0.8816 \pm 0.0011$ | $0.7762 \pm 0.0079$ | $0.8217 \pm 0.0020$ | $0.9011 \pm 0.0031$ | $0.7642 \pm 0.0132$ |
| *scDiff* | $0.9670 \pm 0.0042$ | $0.9680 \pm 0.0143$ | $0.8931 \pm 0.0070$ | $0.8442 \pm 0.0076$ | $0.9473 \pm 0.0074$ | $0.8439 \pm 0.0042$ |

and PBMC1K, from Hou et al. (2020). For evaluation purposes, existing works typically create the corrupted matrix by dropping out some non-zero entries. Specifically, we follow the setting of DCA (Eraslan et al., 2019) and MAGIC (Van Dijk et al., 2018) to mimic the dropout events by masking 10% of the none-zero counts to zeros, where the masking probability is given by the exponential distribution. We evaluate the imputation performance based on the Pearson correlation between the prediction and ground truth on masked entries.

**Baselines.** We choose the following top-performing baselines according to Hou et al. (2020). DCA (Eraslan et al., 2019) parameterizes negative binomial distribution with an autoencoder. MAGIC (Van Dijk et al., 2018) is a graph imputation method based on Markov affinity. ALRA (Linderman et al., 2022) imputes zero counts with low-rank matrix approximation. scVI (Lopez et al., 2018) utilizes a variational autoencoder to approximate negative binomial distribution with cell conditions. SAVER (Huang et al., 2018) estimates the parameters of negative binomial distribution with Poisson LASSO regression. All the baselines are implemented with default settings.

**Experimental results.** The results are illustrated in Table 2a with standard deviation across five runs, where the best result in each dataset is highlighted in bold. We observe that *scDiff* and MAGIC bring out similar performance on Jurkat and 293T, while *scDiff* outperforms the others on PBMC1K. Meanwhile, *scDiff* delivers significant performance gain compared to the other generative model scVI. This highlights the capability of *scDiff* to recover the true expression from dropout events.

Table 2: Results of imputation and perturbation prediction.

(a) Pearson correlation of imputation.

|  | Jurkat | 293T | PBMC1K |
|---|---|---|---|
| ALRA | $0.8070 \pm 0.0001$ | $0.7862 \pm 0.0001$ | $0.6693 \pm 0.0003$ |
| SAVER | $0.7898 \pm 0.0002$ | $0.7735 \pm 0.0003$ | $0.7143 \pm 0.0009$ |
| MAGIC | $0.8249 \pm 0.0001$ | $0.8103 \pm 0.0001$ | $0.7693 \pm 0.0001$ |
| scVI | $0.7285 \pm 0.0013$ | $0.7149 \pm 0.0007$ | $0.6222 \pm 0.0013$ |
| DCA | $0.8189 \pm 0.0002$ | $0.8040 \pm 0.0002$ | $0.7547 \pm 0.0007$ |
| *scDiff* | $0.8228 \pm 0.0001$ | $0.8110 \pm 0.0002$ | $0.7743 \pm 0.0003$ |

(b) Squared Pearson correlation of change in expression on top 100 differential expressed genes.

|  | Salmonella | H.poly | PBMC |
|---|---|---|---|
| CPA | $0.0756 \pm 0.0558$ | $0.2582 \pm 0.1562$ | $0.0742 \pm 0.0235$ |
| Vec | $0.8777 \pm 0.0081$ | $0.8606 \pm 0.0063$ | $0.8346 \pm 0.0008$ |
| PCA-Vec | $0.8752 \pm 0.0058$ | $0.8615 \pm 0.0050$ | $0.8350 \pm 0.0007$ |
| CVAE | $0.8605 \pm 0.0095$ | $0.8964 \pm 0.0146$ | $0.9194 \pm 0.0149$ |
| scGen | $0.8568 \pm 0.0186$ | $0.9115 \pm 0.0060$ | $0.9271 \pm 0.0098$ |
| *scDiff* | $0.9318 \pm 0.0074$ | $0.9169 \pm 0.0160$ | $0.9435 \pm 0.0155$ |

### 4.1.3 PERTURBATION PREDICTION FOR NOVEL CELL TYPE

**Experimental settings.** Previous tasks intend to assess how well the model can approximate the posterior given conditions. On the contrary, the perturbation prediction task requires the model to generalize to novel combinations of conditions. Following scGen (Lotfollahi et al., 2019), we include three datastes, i.e., Salmonella (Haber et al., 2017), H.poly (Haber et al., 2017) and PBMC Kang et al. (2018). Each dataset contains eight different cell types and two perturbation states (perturbed and control). During training, the model is given the full dataset except for the perturbed cells of one cell type, which are held out for testing. Then, the model generates the unseen perturbed cells' expressions for the held-out cell type in the testing stage. The evaluation metric is based on changes in expression between perturbed and control cells. We aggregate the expression of a given condition by calculating the mean expression across cells. The changes in expression are given by the difference between perturbed and control of the held-out cell type in the mean expression. Squared Pearson correlation is calculated among the top 100 differential expressed genes.

**Baselines.** For evaluation purposes, we implement existing baselines and benchmark their performance. scGen (Lotfollahi et al., 2019) is a variational autoencoder combined with latent space vector arithmetics. Along with scGen, we also include some baselines mentioned in Lotfollahi et al. (2019), i.e., a conditional variational autoencoder (CVAE), vector arithmetics on expression space (Vec) and

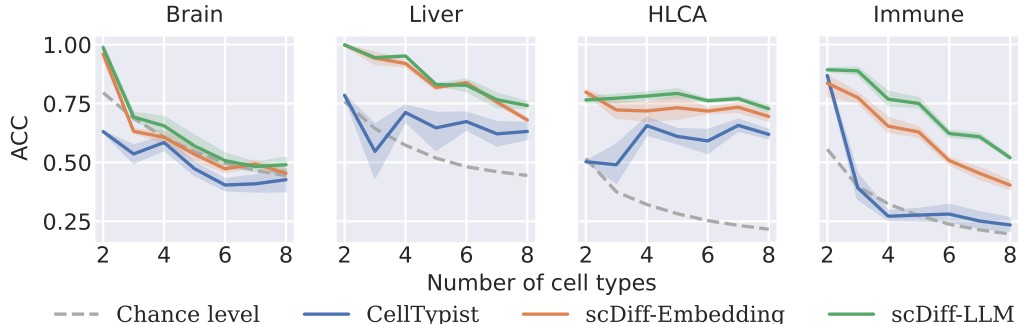

Figure 2: Macro accuracy score of one-shot cell type annotation.

vector arithmetics in the latent space of principal component analyses (PCA-Vec). In addition, we include CPA (Lotfollahi et al., 2023), which is an autoencoder with disentangled latent space.

**Experimental results.** We summarize the results in Table 2b, where the bold numbers represent the top performance of each dataset. Remarkably, *scDiff* outperforms all baselines on all datasets. These results highlight that *scDiff* shows superior generalizability to novel conditions compared to the baseline counterparts. Consequently, *scDiff* shows great potential in single-cell applications where condition transfer is needed.

## 4.2 PERFORMANCE OF EXTERNAL-CONDITIONED *scDiff*

In Section 4.1, we evaluated *scDiff* in three representative tasks where all individual conditions are observed. In practice, we may encounter extreme cases when only a few or even no labeled samples are available for the query conditions. Given only the internal information, the few-shot or zero-shot settings are challenging, if not intractable. Here, we showcase ways to extend *scDiff* by incorporating prior information as external conditions to enable the handling of unseen conditions. To test the performance of *scDiff* under these settings, we conduct experiments with one-shot cell type annotation and zero-shot gene perturbation prediction.

### 4.2.1 ONE-SHOT CELL TYPE ANNOTATION

While some rare cell types play a crucial role in particular researches (Khalilia et al., 2011), accurately annotating them is incredibly challenging because of the limited availability of labeled samples (Jindal et al., 2018). Under the few-shot setting, prior information on cell types would significantly enhance the model. The cell ontology provides a comprehensive vocabulary and definitions of different cell types written in natural language, which can be readily encoded by LLMs into embeddings. We use BioLinkBERT (Yasunaga et al., 2022) as the backbone LLM since it is specifically trained on the biomedical corpuses. We extracted textual descriptions of all cell types from the cell ontology terms that appeared in our datasets except for the mucus-secreting cell (CL:0000319) and pulmonary artery endothelial cell (CL:1001568). For these two terms, we utilize GPT4 (OpenAI, 2023) to depict their descriptions given available definitions as query contexts.

**Experimental settings.** For the six datasets in Section 4.1.1, we use the four datasets from CELLxGENE (Megill et al., 2021) since they come with manually annotated cell ontology terms. To mimic the rare samples setting, we simulate an extreme one-shot scenario. Particularly, for a specific dataset, we refer to the cell types that contain more cells than a threshold as majority types and the remaining as minority types. We randomly sample one cell per type from the minority types as the one-shot set, and the rest of the cells in minority types are used for evaluation. *scDiff* is first pretrained on all cells in majority types and then fine-tuned on the one-shot set for 50 epochs. Similar to the evaluation in Section 4.1.1, we calculate the macro average of the multi-class accuracy score and F1 score for performance comparison. More details can be found in Appendix E.4.

**Experimental results.** We summarize the results in Fig. 2, where we increase the number of cell types by gradually adding target cell types from top to bottom in the descending order of their original cell counts. We directly train a CellTypist (Domínguez Conde et al., 2022) on the one-shot set to provide a reference of the macro accuracy scores. Notably, the LLM variant of *scDiff* shows performance gain against the class-conditioned *scDiff* on three out of four datasets. The conclusion can be drawn from the results that utilizing BioLinkBERT in *scDiff*-LLM enhances the model in the one-shot setting. More results and analysis are in Appendix E.4.

### 4.2.2 NOVEL GENE PERTURBATION PREDICTION

Understanding the transcriptional responses to genetic perturbations is a crucial step towards delineating the regulatory circuits in the biological system (Jaitin et al., 2016; Sachs et al., 2005). Its realization has many vital applications in translational medicine and health science (Réda et al., 2020). However, exhaustively screening all possible genetic perturbations is impractical, given the high cost of such experiments. Here, we follow GEARS (Roohani et al., 2023) and leverage *scDiff* to predict the effects of novel gene perturbations. Under such zero-shot setting, we incorporate biological prior for the unseen genes by adapting the gene ontology-based graph neural network from GEARS, as detailed in Section 2.2.

**Experimental settings.** To access the zero-shot gene perturbation prediction performance, we follow the setting of GEARS by holding out part of the perturbations as the test set. We include two one-gene perturbation datasets (Adamson et al., 2016; Dixit et al., 2016) and a two-gene perturbation dataset (Norman et al., 2019). To be consistent with GEARS, we use the same metrics, i.e., Pearson correlation of change in expression (Delta Pearson Correlation) and mean squared error of change in expression on top 20 differential expressed genes (MSE Top 20 DE).

**Experimental results.** The results are illustrated in Fig. 3. We observe that *scDiff* outperforms GEARS among all the metrics and datasets except the MSE on Norman[3]. In addition, *scDiff* presents more stable results with smaller variance across five runs. This observation indicates that the core components of GEARS can be readily adapted to *scDiff* as a conditioner without any modification. The numeric results are summarized in Appendix E.5.

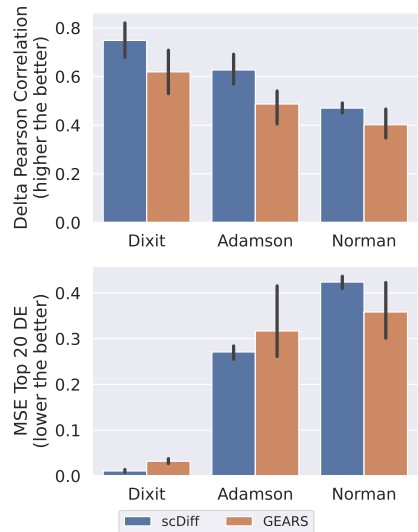

Figure 3: Gene perturbation performance comparison.

## 5 CONCLUSION

In this work, we have unified common single-cell tasks with a posterior modeling framework. Subsequently, we developed *scDiff* using a conditional diffusion generative model to approximate the posterior. *scDiff* showed decent performances in diverse single-cell benchmarking tasks using a single training objective. More importantly, the proposed *scDiff* is versatile and accommodates various conditioning strategies. As two showcases, we incorporated prior information with large language models and graph neural networks. Our results demonstrated that *scDiff* successfully leveraged this prior information through conditioning. Together, our work paves the way for diffusion generative models in single-cell analysis, ultimately accelerating the development from health science to therapeutic discovery.

**Future work and limitation.** The flexibility of *scDiff* enables extensive conditioning strategies. Besides LLMs and GNNs, we can enhance *scDiff* with other guidance methods, like CLIP (Radford et al., 2021; Kim et al., 2022b). In addition, the proposed probabilistic framework can be promptly extended to multiomics or multi-modality tasks. A natural future direction is to explore the possibility of *scDiff* in spatial transcriptomic. There, histological images can also be used as an additional condition, further opening up the possibilities in single cell analyses such as predicting gene expression from histology (Shmatko et al., 2022). On the other hand, there are intrinsic limitations of the current framework. Particularly, representation learning plays a crucial role in several single-cell tasks, such as single-cell integration (Luecken et al., 2022). Yet, *scDiff* at its current form still needs further adaptation to accommodate those tasks. With recent works in vision (Preechakul et al., 2022; Kim et al., 2022a), a promising future direction of *scDiff* is meaningful representation learning.

---

[3]We directly copied GEARS' performance on the Dixit dataset from the manuscript due to reproducibility issues (Appendix E.5)

## 6 ETHICAL STATEMENT

All data used in this work are from public resources and thus adhere to their license agreements. We confirm that no known privacy and ethics issues are associated with the public data we chose to use.

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

## A   FURTHER DETAILS OF SINGLE-CELL TASKS

**Cell type deconvolution** is to estimate the proportions of different cell types within mixed-cell spatial transcriptomics data. Due to the nature of the spatial transcriptomics profiling technologies, the true cell type compositions are unknown. This task typically requires some external reference, in most cases using annotated scRNA-seq data as a reference (Biancalani et al., 2021; Cable et al., 2022; Ma & Zhou, 2022).

**Modality prediction** aims to estimate expression levels of target modality from the input modality. As in equation 2, let $\mathbf{X} = [\mathbf{X}_{\text{source}}, \mathbf{X}_{\text{target}}]$ be the concatenation of measures of source modality $\mathbf{X}_{\text{source}} \in \mathbb{R}^{n,m_s}$ and target modality $\mathbf{X}_{\text{target}} \in \mathbb{R}^{n,m_t}$, then the indicator matrix can be written as $\mathbf{M} = [\mathbf{J}_{n,m_s}, \mathbf{O}_{n,m_t}]$, where $m_s$ and $m_t$ are the number of source genes and target genes. Note that for ease of notation, we present the above formulation assuming that the two modalities have the same number of cells, which can be readily satisfied by appending all-zeros samples.

## B   PARAMETERIZATION DETAILS

As described in DDPM (Ho et al., 2020), the ELBO in equation 8 can be written as:

$$\mathbb{E}_q[\underbrace{D_{\text{KL}}\left(q\left(\mathbf{x}_T \mid \mathbf{x}_0\right) \| p\left(\mathbf{x}_T\right)\right)}_{L_T} + \sum_{t>1}\underbrace{D_{\text{KL}}\left(q\left(\mathbf{x}_{t-1} \mid \mathbf{x}_t, \mathbf{x}_0\right) \| p_\theta\left(\mathbf{x}_{t-1} \mid \mathbf{x}_t, \mathbf{c}\right)\right)}_{L_{t-1}} \underbrace{- \log p_\theta\left(\mathbf{x}_0 \mid \mathbf{x}_1, \mathbf{c}\right)}_{L_0}].$$

$$(15)$$

Following Rombach et al. (2022), the posterior mean in equation 7 has the form:

$$\boldsymbol{\mu}_\theta\left(\mathbf{x}_t, t\right) = \frac{\sqrt{\bar{\alpha}_t}(1 - \bar{\alpha}_{t-1})}{\sqrt{\bar{\alpha}_{t-1}}(1 - \bar{\alpha}_t)}\mathbf{x}_t + \frac{\sqrt{\bar{\alpha}_t}(1 - \frac{\bar{\alpha}_{t-1}}{\bar{\alpha}_{t-1}})}{1 - \bar{\alpha}_t}\mathbf{x}_\theta\left(\mathbf{x}_t, t\right). \tag{16}$$

Replacing $\mathbf{x}_t$ with $\sqrt{\bar{\alpha}_t}\mathbf{x}_0 + \sqrt{1 - \bar{\alpha}_t}\boldsymbol{\epsilon}$, the KL divergence term $L_{t-1}$ simplifies to:

$$\mathbb{E}_{\mathbf{x}_0,\boldsymbol{\epsilon}}\left[\frac{1}{2}(\frac{\bar{\alpha}_{t-1}}{1 - \bar{\alpha}_{t-1}} - \frac{\bar{\alpha}_t}{1 - \bar{\alpha}_t})\left\|\mathbf{x}_0 - \mathbf{x}_\theta\left(\sqrt{\bar{\alpha}_t}\mathbf{x}_0 + \sqrt{1 - \bar{\alpha}_t}\boldsymbol{\epsilon}, \mathbf{c}, t\right)\right\|^2\right]. \tag{17}$$

Since the term $L_T$ has no trainable parameters, the complete training objective becomes:

$$\sum_{t=1}^T \mathbb{E}_{\mathbf{x}_0,\boldsymbol{\epsilon}}\left[\frac{1}{2}(\frac{\bar{\alpha}_{t-1}}{1 - \bar{\alpha}_{t-1}} - \frac{\bar{\alpha}_t}{1 - \bar{\alpha}_t})\left\|\mathbf{x}_0 - \mathbf{x}_\theta\left(\sqrt{\bar{\alpha}_t}\mathbf{x}_0 + \sqrt{1 - \bar{\alpha}_t}\boldsymbol{\epsilon}, \mathbf{c}, t\right)\right\|^2\right]. \tag{18}$$

Apply the simplification process described in DDPM, the training objective reduces to equation 9 by ignoring the weights of every time-step.

## C   FURTHER DETAILS OF *scDiff*

### C.1   SINUSOIDAL TIME EMBEDDING

In equation 10, we implement sinusoidal time embedding as:

$$\text{TimeEmbed}_{[t,i]} = \begin{cases} \sin\left((t/T)^{2i/d}\right) & \text{if } 1 \le i \le \lfloor d/2 \rfloor \\ \cos\left((t/T)^{2i/d}\right) & \text{otherwise} \end{cases} \tag{19}$$

### C.2   CONTEXT EMBEDDING PROCESSING

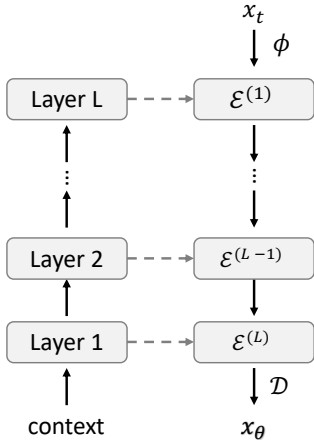

Figure 4: Illustration of context conditioner.

As illustrated in Fig. 4, unlike the parallel processing done with parallel MLP for other conditioners mentioned in section 2.2, the masked gene expression context conditioner $\psi_{f_{\text{ctxt}}}$ extracts the hidden representations within a single MLP. Formally,

$$\psi_{f_{\text{ctxt}}}^{(l)}(\tilde{\mathbf{x}}) = \text{MLP}_{\text{ctxt}}^{[1:(L-l+1)]}(\tilde{\mathbf{x}}), \tag{20}$$

where $\text{MLP}_{\text{ctxt}}^{[1:(L-l+1)]}$ is the output of the MLP at the $l^{\text{th}}$ last layer. Subsequently, the processed condition embeddings are fed into the cross-attention blocks in reverse, where the highly processed context embeddings are first mixed with the raw input embeddings. This reversed mixing approach is inspired by the similar design of the DiffMAE model (Wei et al., 2023), where the masked and visual patches are mixed in reversed order. We found that, empirically, using the reversed mixing strategy leads to better reconstruction of the gene expression ($\mathbf{x}_0$ predictions) during training, as opposed to using the multihead MLP processing.

### C.3   CROSS-ATTENTION

We specify the cross-attention in equation 12 as:

$$\text{CrossAttn}(\mathbf{h}, \{\mathbf{z}_i\}_i) = \text{FFN}\Big( \sum_i \frac{\exp(\mathbf{h}\mathbf{W}_k\mathbf{z}_i^\top)}{\sum_j \exp(\mathbf{h}\mathbf{W}_k\mathbf{z}_j^\top)} \mathbf{z}_i\mathbf{W}_v \Big). \tag{21}$$

The feed-forward network (FFN) can be formulated as:

$$\text{FFN}(\mathbf{z}) = \text{GELU}\big(\text{LayerNorm}(\mathbf{z})\mathbf{W}_1 + \mathbf{b}_1\big)\mathbf{W}_2 + \mathbf{b} + \mathbf{z}. \tag{22}$$

## D   TASK-AGNOSTIC TRAINING AND TASK-SPECIFIC INFERENCE

In Section 2.1, we formulate all the tasks as posterior modeling questions. Subsequently, *scDiff* can handle multiple downstream tasks with the same trained model through diverse inference processes,

provided that the required condition type is embedded in the model. This allows us to unify the training process of all the tasks by consolidating all conditions into the prior. While the available conditions are determined by the datasets rather than the tasks, we term the training process of *scDiff* as data-centric training, or equivalently, task-agnostic training. Specifically, for a given dataset, we gather all conditions, including cell types, cell batch labels, cell context, and, if available, perturbation status. Along with diffused expression, all the conditions are then fed into *scDiff* to recover the clean expression with the training objective in equation 9. In the following sections, we present the details of the task-specific inference stage.

## D.1 CELL TYPE ANNOTATION

Li et al. (2023) introduced a density estimation approach to probe the class prediction of an input sample out of the diffusion model. The core idea of the method lies in evaluating the prediction error of the input $\mathbf{x}_0$ at various time steps $t$ given different conditions $\mathbf{c}$. Consequently, the condition that leads to the smallest noise prediction error is taken as the prediction of the input sample. Here, we adapt this approach to the predict-$\mathbf{x}_0$ scenario. Recall that the posterior for a discrete class variable $\mathbf{c}$ can be described as

$$p_\theta\left(\mathbf{c}_i \mid \mathbf{x}\right) = \frac{p\left(\mathbf{c}_i\right) p_\theta\left(\mathbf{x} \mid \mathbf{c}_i\right)}{\sum_j p\left(\mathbf{c}_j\right) p_\theta\left(\mathbf{x} \mid \mathbf{c}_j\right)} = \frac{p_\theta\left(\mathbf{x} \mid \mathbf{c}_i\right)}{\sum_j p_\theta\left(\mathbf{x} \mid \mathbf{c}_j\right)}, \tag{23}$$

assuming a uniform prior over $\{\mathbf{c}_i\}$. Since the data posterior is intractable for the diffusion models, we replace $p_\theta\left(\mathbf{x} \mid \mathbf{c}_i\right)$ with the exponential of negative simplified training objective

$$p_\theta\left(\mathbf{c}_i \mid \mathbf{x}\right) = \frac{\exp\left\{-\mathbb{E}_{t,\mathbf{x}_0,\epsilon}\left[\left\|\mathbf{x}_0 - \mathbf{x}_\theta\left(\sqrt{\bar{\alpha}_t}\mathbf{x}_0 + \sqrt{1-\bar{\alpha}_t}\epsilon, \mathbf{c}_i\right)\right\|^2\right]\right\}}{\sum_j \exp\left\{-\mathbb{E}_{t,\mathbf{x}_0,\epsilon}\left[\left\|\mathbf{x}_0 - \mathbf{x}_\theta\left(\sqrt{\bar{\alpha}_t}\mathbf{x}_0 + \sqrt{1-\bar{\alpha}_t}\epsilon, \mathbf{c}_j\right)\right\|^2\right]\right\}}. \tag{24}$$

An unbiased estimation of the simplified training objective can be obtained by Monte Carlo estimation. This can be done by repeatedly sampling $t$ from uniform distribution $\mathcal{U}[1, 1000]$, $\epsilon$ from standard Gaussian, and calculating

$$\sum_{i=1}^{N} \left\|\mathbf{x}_0 - \mathbf{x}_\theta\left(\sqrt{\bar{\alpha}_t}\mathbf{x}_0 + \sqrt{1-\bar{\alpha}_t}\epsilon, \mathbf{c}_i\right)\right\|^2, \tag{25}$$

where $N$ is the number of Monte Carlo samples. In practice, we follow Li et al. (2023) and use 5 equally spaced-out time-steps between 0 and $T$.

## D.2 MISSING VALUE IMPUTATION

In the imputation setting, it is important to note that there is a gap between the generated samples and the missing values. We incorporate cell context as a condition during the training stage to model the posterior of the training data. In the inference stage, we approximate the missing values by sampling from the distribution of training data conditioned on the visible values. Empirically we find out that this is a good approximation of the imputation task.

## D.3 PERTURBATION PREDICTION

We apply a similar inference strategy in perturbation prediction tasks for novel cell types and novel gene perturbations. In both scenarios, *scDiff* is trained on the existing conditions in the training set, and the task objective is to generate samples from query conditions.

In the case of novel cell types, we aim to predict the expression of a hold-out cell type under perturbation. During the training stage, *scDiff* has seen the perturbed state of other cell types. The query condition in the testing set is the combination of held-out cell type and positive perturbation status.

In the context of novel gene perturbation, the query conditions involve novel types of gene perturbations that are excluded from the training set. In such a zero-shot scenario, we incorporate GEARS (Roohani et al., 2023) conditioner to provide prior information about the query conditions for *scDiff*.

# E    EXPERIMENTAL SETTINGS AND RESULTS

## E.1    IMPLEMENTATION DETAILS

We used a unified *scDiff* model architecture throughout the paper for different tasks. We train a new model from scratch for each task on each dataset except for the few-shot annotation setting. The only changing parts across tasks are the number of epochs, and the base learning rate of the optimizer (Table 3). These two choices are adjusted according to the sizes of the particular dataset.

**Architecture** We used six layers of cross-attention blocks to encode the cell latent representations. Accordingly, each conditioner $\psi_f$ is a collection of six MLPs. We set the latent dimension to $512$ and used $8$ stacking attention heads. More specifically, each attention head is $64$ dimensional, and we combine outputs from all the attention heads by concatenating them into a $512$ dimensional vector.

**Optimizer** We used the decoupled Adam (AdamW) (Loshchilov & Hutter, 2018) to optimizer the model's parameter. The learning rate is computed by scaling the base learning rate with the batch size, learning_rate = base_learning_rate × batch_size. We use batch size of 2048 throughout the experiments.

Table 3: Task specific hyperparameter settings and the choice of conditioners enabled. † indicates that the LLM conditioner can be optionally turned on to replace the cell-type class conditioner.

| | | Max epochs | Base Learning Rate | Context | Cell type | Conditiners Perturb state | Gene perturbation | LLM |
|---|---|---|---|---|---|---|---|---|
| Denoising | Jurkat | 3000 | 1.00e-8 | ✓ | ✓ | | | |
| | 293T | 3000 | 1.00e-8 | ✓ | ✓ | | | |
| | PBMC1K | 3000 | 1.00e-8 | ✓ | ✓ | | | |
| Annotation | PBMC12K | 2000 | 1.00e-8 | ✓ | ✓ | | | |
| | Pancreas | 2000 | 1.00e-8 | ✓ | ✓ | | | |
| | HLCA (subset) | 2000 | 1.00e-8 | ✓ | ✓ | | | † |
| | Immune (subset) | 2000 | 1.00e-7 | ✓ | ✓ | | | † |
| | Brain | 2000 | 1.00e-8 | ✓ | ✓ | | | † |
| | Liver | 2000 | 1.00e-8 | ✓ | ✓ | | | † |
| Perturbation | Salmonella | 1000 | 1.00e-8 | ✓ | ✓ | ✓ | | |
| | HPoly | 1000 | 1.00e-8 | ✓ | ✓ | ✓ | | |
| | PBMC | 1000 | 5.00e-9 | ✓ | ✓ | ✓ | | |
| Gene Pert | Adamson | 50 | 1.00e-8 | ✓ | ✓ | | ✓ | |
| | Norman | 100 | 1.00e-8 | ✓ | ✓ | | ✓ | |
| | Dixit | 150 | 1.00e-8 | ✓ | ✓ | | ✓ | |

## E.2    DATASETS

For all the datasets, we applied the standard preprocessing pipeline: filtering out all-zero genes and cells, library size normalization with target sum $= 10,000$, and then logarithm normalization. We summarize the information of all processed datasets in Table 4.

Table 4: Summary of datasets

| | Dataset | # cells | # genes | # batchs | # cell types | # conditions | Reference | Link |
|---|---|---|---|---|---|---|---|---|
| Denoising | Jurkat | 3,258 | 12,635 | – | – | – | Zheng et al. (2017); Hou et al. (2020) | 10x Genomics |
| | 293T | 2,885 | 13,555 | – | – | – | Zheng et al. (2017); Hou et al. (2020) | 10x Genomics |
| | PBMC1K | 1,087 | 12,811 | – | – | – | Hou et al. (2020) | 10x Genomics |
| Annotation | PBMC12K | 11,990 | 3,254 | 2 | 9 | – | Zheng et al. (2017); Lopez et al. (2018) | scvi-tools |
| | Pancreas | 16,382 | 16,587 | 9 | 14 | – | Luecken et al. (2022) | Open Problems |
| | HLCA (subset) | 58,484 | 28,024 | 166 | 47 | – | Sikkema et al. (2023) | CELLxGENE |
| | Immune (subset) | 32,964 | 20,661 | 12 | 33 | – | Domínguez Conde et al. (2022) | CELLxGENE |
| | Brain | 45,528 | 26,469 | 20 | 14 | – | Seeker et al. (2023) | CELLxGENE |
| | Liver | 8,444 | 22,461 | 5 | 13 | – | MacParland et al. (2018) | CELLxGENE |
| Perturbation | Salmonella | 5,010 | 15,215 | 2 | 8 | 2 | Lotfollahi et al. (2019); Haber et al. (2017) | GSE92332 |
| | HPoly | 5,951 | 15,215 | 2 | 8 | 2 | Lotfollahi et al. (2019); Haber et al. (2017) | GSE92332 |
| | PBMC | 17,082 | 15,109 | 3 | 8 | 2 | Lotfollahi et al. (2019); Kang et al. (2018) | GSE96583 |
| Gene Perturbation | Dixit | 44,735 | 5,012 | – | – | 20 | Roohani et al. (2023); Dixit et al. (2016) | GSE90063 |
| | Adamson | 68,603 | 5,060 | – | – | 87 | Roohani et al. (2023); Adamson et al. (2016) | GSE90546 |
| | Norman | 91,205 | 5,045 | – | – | 284 | Roohani et al. (2023); Norman et al. (2019) | GSE133344 |

### E.3    CELL TYPE ANNOTATION

We present the macro F1 scores of the cell-type annotation task in Tabke 5. It implies the same conclusion that *scDiff* outperforms the baselines in four out of six datasets.

Table 5: Macro F1 score of cell type annotation

|  | PBMC12K | Pancreas | HLCA | Immune | Brain | Liver |
|---|---|---|---|---|---|---|
| CellTypist | **0.9081 +/- 0.0029** | **0.8946 +/- 0.0013** | **0.8253 +/- 0.0068** | **0.8523 +/- 0.0013** | **0.9229 +/- 0.0023** | 0.8001 +/- 0.0080 |
| SingleCellNet | 0.8789 +/- 0.0062 | 0.6489 +/- 0.0006 | **0.8449 +/- 0.0052** | 0.8045 +/- 0.0008 | **0.9047 +/- 0.0032** | **0.8889 +/- 0.0020** |
| ACTINN | 0.6243 +/- 0.0790 | 0.5307 +/- 0.1023 | 0.1955 +/- 0.0456 | 0.1988 +/- 0.0370 | 0.6865 +/- 0.0760 | 0.6174 +/- 0.0374 |
| scANVI | **0.9107 +/- 0.0089** | **0.8738 +/- 0.0269** | 0.6119 +/- 0.2285 | **0.8227 +/- 0.0204** | 0.8094 +/- 0.0152 | **0.8352 +/- 0.0131** |
| *scDiff* | **0.9485 +/- 0.0062** | **0.9626 +/- 0.0089** | **0.8949 +/- 0.0057** | **0.8193 +/- 0.0106** | **0.9534 +/- 0.0065** | **0.8663 +/- 0.0100** |

### E.4    FEW-SHOT CELL TYPE ANNOTATION

Table 6: Dataset statistics of few-shot annotation

|  | threshold | # pre-train cell types | # pre-train cells |
|---|---|---|---|
| Brain | 3000 | 6 | 43750 |
| Liver | 600 | 5 | 4613 |
| HLCA | 1000 | 14 | 49279 |
| Immune | 1000 | 14 | 16969 |

On each dataset, we split the set of cells $\mathbb{C}$ into two sets: majority set $\mathbb{C}_{majority}$ and minority set $\mathbb{C}_{minority}$. The majority set $\mathbb{C}_{majority}$ consists of cells from the majority cell types that contain more cells than the threshold, while the remaining cells form the minority set $\mathbb{C}_{minority}$. In the one-shot setting, we randomly sample one cell from each type in $\mathbb{C}_{minority}$ to form the one-shot set $\mathbb{C}_{one}$. *scDiff* is pre-trained on $\mathbb{C}_{majority}$ for 1000 epochs with the base learning rate as $1e - 8$ and fine-tuned on $\mathbb{C}_{one}$ for 50 epochs, as mentioned in Section 4.2.1. We evaluate the performance of *scDiff* on the remaining part of minority set $\mathbb{C}_{minority} - \mathbb{C}_{one}$. Similarly, for the few-shot setting, we increase the number of cells per type sampled from the minority set $\mathbb{C}_{minority}$. We summarize the statistics of the datasets used in few-shot cell type annotation in Table 6.

We illustrate the macro F1 score of one-shot cell type annotation in Fig. 5. Note that there is a significant drop of performance when the number of cell types reaches 5 in Liver dataset. The fifth cell type is B cell, and we observe that plasma cell have been included as the third cell type. Since B cells and plasma cells are from lymphocyte of B lineage (Hoffman et al., 2016), their gene expression levels are similar to each other. In the one-shot setting, two cells from different cell types share similar expression measurements, which will obscure the model. This observation reveals one drawback of diffusion classifier: it becomes increasingly challenging for diffusion classifier when the distributions of different classes are indistinguishable.

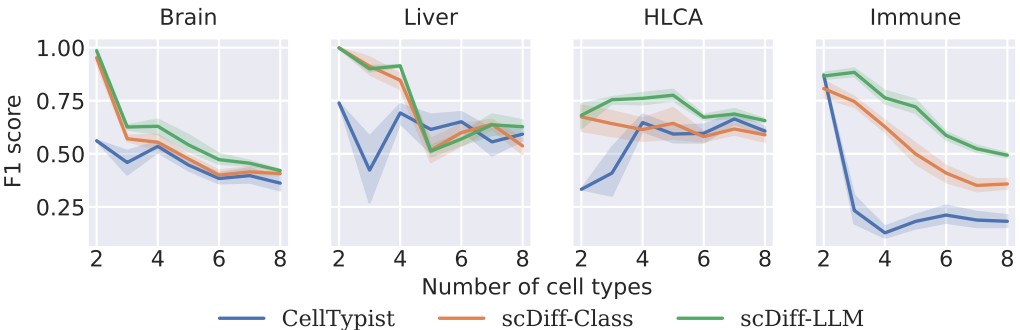

Figure 5: Macro F1 score of one-shot cell type annotation.

We summarize the macro accuracy score of few-shot cell type annotation in Fig. 6 and the macro F1 score in Fig. 7. We fix the number of target cell types to 5. In Brain and Immune, LLM-guided

*scDiff* shows significant performance gain against class-guided *scDiff*. In the other two datasets, the two variants of *scDiff* achieve comaprable performance.

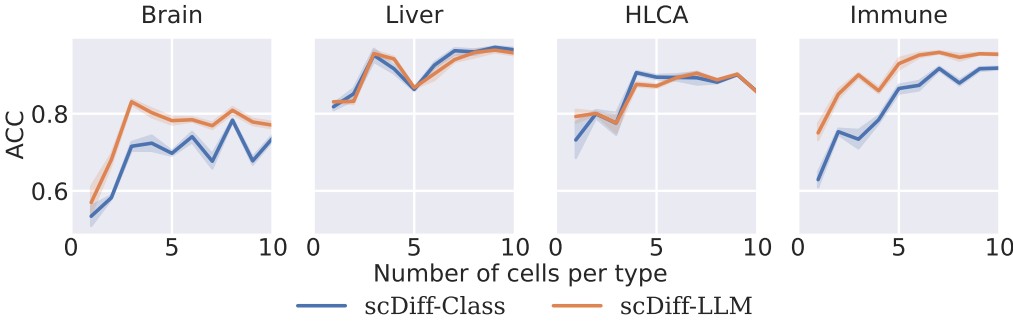

Figure 6: Macro accuracy score of few-shot cell type annotation on top $5$ cell types.

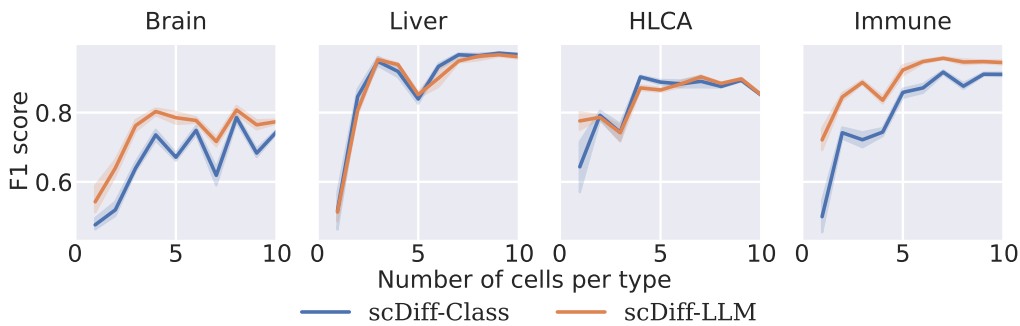

Figure 7: Macro F1 score of few-shot cell type annotation on top $5$ cell types.

### E.5 NOVEL GENE PERTURBATION PREDICTION

Table 7: Gene perturbation performance comparison.

|  | Dataset | Adamson | Dixit | Norman |
|---|---|---|---|---|
| GEARS | Corr Delt ↑ | 0.4869 +/- 0.0890 | 0.6191 +/- 0.1156 | 0.4018 +/- 0.0719 |
|  | MSE DE 20 ↓ | 0.3169 +/- 0.1103 | 0.0319 +/- 0.0068 | 0.3583 +/- 0.0802 |
| *scDiff* | Corr Delt ↑ | 0.6269 +/- 0.0820 | 0.7483 +/- 0.0922 | 0.4702 +/- 0.0280 |
|  | MSE DE 20 ↓ | 0.2709 +/- 0.0187 | 0.0106 +/- 0.0041 | 0.4238 +/- 0.0175 |

We used the official GEARS code base[4] to reproduce the performance on the three benchmarking datasets. We need to rerun the experiments because the data splits used from the original paper were not published. Thus, directly comparing our results with the reported metrics from the GEARS paper is infeasible. We obtained results comparable with the reported scores from the paper for the Adamson and Norman datasets. However, we cannot produce reasonable results for the Dixit dataset and have obtained negative Pearson correlations. Thus, we decided to directly copy the reported metrics for Dixit rather than using the doubtful results.

### E.6 ABLATION STUDIES

We present the ablation studies in the annotation task regarding the number of layers in the embedder and decoder, adding residual connection and removing cross-attention. The results are summarized

---

[4]https://github.com/yhr91/GEARS_misc/tree/6d34e646fc878f897625dc4d5cc10c30234aace4

Table 8: Macro accuracy score of ablation studies on annotation.

| | | Brain | HLCA | Immune | Liver | Pancreas | PBMC12K |
|---|---|---|---|---|---|---|---|
| | Default | 0.9473 +/- 0.0074 | 0.8931 +/- 0.0070 | 0.8442 +/- 0.0076 | 0.8439 +/- 0.0042 | 0.9680 +/- 0.0143 | 0.9670 +/- 0.0042 |
| Decoder | 2-layer | 0.9445 +/- 0.0041 | 0.8911 +/- 0.0042 | 0.8444 +/- 0.0070 | 0.8589 +/- 0.0238 | 0.9735 +/- 0.0023 | 0.9700 +/- 0.0045 |
| | 3-layer | 0.9378 +/- 0.0017 | 0.8994 +/- 0.0038 | 0.8383 +/- 0.0112 | 0.8062 +/- 0.0216 | 0.9731 +/- 0.0014 | 0.9701 +/- 0.0029 |
| | 4-layer | 0.9409 +/- 0.0064 | 0.8989 +/- 0.0037 | 0.8414 +/- 0.0051 | 0.8199 +/- 0.0185 | 0.9763 +/- 0.0011 | 0.9744 +/- 0.0002 |
| Embedder | 2-layer | 0.9394 +/- 0.0029 | 0.8857 +/- 0.0089 | 0.8424 +/- 0.0061 | 0.8434 +/- 0.0239 | 0.9720 +/- 0.0011 | 0.9667 +/- 0.0039 |
| | 3-layer | 0.9512 +/- 0.0048 | 0.8847 +/- 0.0053 | 0.8358 +/- 0.0081 | 0.8447 +/- 0.0224 | 0.9745 +/- 0.0026 | 0.9672 +/- 0.0039 |
| | 4-layer | 0.9434 +/- 0.0048 | 0.8927 +/- 0.0018 | 0.8461 +/- 0.0014 | 0.8423 +/- 0.0100 | 0.9742 +/- 0.0012 | 0.9606 +/- 0.0036 |
| General | U-Net residual | 0.9456 +/- 0.0047 | 0.8959 +/- 0.0050 | 0.8502 +/- 0.0075 | 0.8399 +/- 0.0103 | 0.9693 +/- 0.0014 | 0.9644 +/- 0.0021 |
| | No cross attention | 0.9376 +/- 0.0019 | 0.8870 +/- 0.0012 | 0.8318 +/- 0.0028 | 0.8456 +/- 0.0054 | 0.9722 +/- 0.0000 | 0.9608 +/- 0.0042 |
| | Remove diffusion | 0.5513 +/- 0.1390 | 0.1487 +/- 0.0313 | 0.1233 +/- 0.1661 | 0.1525 +/- 0.0214 | 0.3749 +/- 0.0627 | 0.5939 +/- 0.1144 |

Table 9: Pearson correlation of ablation studies on imputation.

| | Jurkat | 293T | PBMC1K |
|---|---|---|---|
| *scDiff* | 0.8228 +/- 0.0001 | 0.8110 +/- 0.0002 | 0.7743 +/- 0.0003 |
| Remove diffusion | 0.7833 +/- 0.0021 | 0.7674 +/- 0.0014 | 0.7439 +/- 0.0004 |

in Table 8, where we report the accuracy across all the annotation datasets. We will elaborate on our motivation for the current model structure according to the results.

The autoencoder structure has shown great success in single-cell analysis (Lopez et al., 2018; Eraslan et al., 2019). Following the existing works (Gong et al., 2023; He et al., 2022), we build our model with a relatively heavy 6-layer encoder and a light 1-layer decoder. Before the encoder, we utilize an embedder to project the input expression into the embedding space. In the encoder, we wish to inject the prior information through different conditions into the expression embeddings, where we choose the cross-attention mechanism due to performance concerns. We extract the embeddings of conditions through various conditioners and feed them into the key and value of the cross-attention blocks. The embedder serves as a projection from expression space to embedding space, while the decoder reverses this process. We empirically observed that a 1-layer neural network is sufficient for such transformation. We note that the model structure has not been optimized for downstream tasks. Tricks like adding U-Net style residual connections can still improve the performance.

We also include the ablation study of removing the diffusion process for annotation and imputation in Table 8 and Table 9, respectively. In this experiment, we retain the model structure in both tasks. Annotation is achieved by selecting the class that minimizes the reconstruction error of the posterior, while we impute the missing values through 1-step prediction from the visible entries. We observe that removing the diffusion process results in a significant performance drop in both tasks, indicating that the diffusion process serves as a crucial part of the proposed probabilistic modeling framework.

## E.7 COMPUTATION EFFICIENCY

Table 10: Per-epoch training run time in seconds.

| | Dataset | *scDiff* | Baseline | Baseline name |
|---|---|---|---|---|
| Annotation | Brain | 9.63 | 9.82 | scANVI |
| | HLCA | 13.6 | 12.73 | scANVI |
| | Immune | 5.28 | 8.86 | scANVI |
| | Liver | 2.83 | 2.97 | scANVI |
| | Pancreas | 5.05 | 14.37 | scANVI |
| | PBMC12K | 2.04 | 3.31 | scANVI |
| Denoising | 293T | 0.8847 | 0.53 | DCA |
| | Jurkat | 0.5736 | 0.6 | DCA |
| | PBMC1K | 0.2634 | 0.16 | DCA |
| Perturbation | HPoly | 1.96 | 1.83 | scGEN |
| | PBMC | 3.16 | 3.53 | scGEN |
| | Salmonella | 1.58 | 1.81 | scGEN |
| Gene Perturbation | Adamson | 83.88 | 1189.66 | GEARS |
| | Dixit | 44.67 | 760.7 | GEARS |
| | Norman | 155.88 | 491.97 | GEARS |

We note that *scDiff* has a computational complexity that is nearly the same as a standard MLP. In Table 10, we report the per-epoch computation time for *scDiff* against standard baselines for all our benchmarking tasks. It is crucial to acknowledge that diffusion models inherently demand a relatively longer time during the inference stage compared to other generative models. In our empirical observations, we find that the inference time of *scDiff* is negligible compared to the training time. Therefore, our emphasis is placed on training speed.

From the table, we observe that in all cases, *scDiff* has comparable per-epoch runtime as the baselines, especially for MLP-based ones (scANVI, DCA, and scGEN). Referring to the model structure of *scDiff*, the central backbone of it comprises the cross-attention (CA) encoder blocks, where the tokens are the number of types of conditions (e.g., cell type, perturbation status). As mentioned in Appendix E.2 Table 4, the number of types of conditions will not exceed 5. Thus, unlike the traditional transformer model that poses $O(N^2)$ complexity, where the token number $N$ is typically much larger than the hidden dimensions, in *scDiff* the feedforward module dominates the computation complexity of our CA block.

For the space complexity, we note that the required amount of GPU memory is determined by both the batch size and the number of genes. Empirically, we set batch_size $= 2048$ throughout all the experiments. On the dataset with the highest number of genes, namely HLCA with $28,024$ genes, *scDiff* necessitates $4.5$ GB of GPU memory. Given the above discussion on training efficiency, it is noteworthy that *scDiff* can be scaled effectively to large datasets.

### E.8 ADDITIONAL BENCHMARKING RESULTS FOLLOWING OPENPROBLEMS

Table 11: Macro accuracy of annotation following OpenProblems

|  | Pancreas | Tabula Muris | CeNGEN |
|---|---|---|---|
| Logistic regression | 0.98 | 0.92 | 0.89 |
| Seurat reference mapping | 0.98 | 0.90 | 0.83 |
| Multilayer perceptron | 0.98 | 0.92 | 0.87 |
| XGBoost | 0.97 | 0.86 | 0.84 |
| *scDiff* | 0.9800 +/- 0.0039 | 0.9257 +/- 0.0066 | 0.8889 +/- 0.0024 |

Table 12: Mean square error of denoising following OpenProblems

|  | PBMC | Pancreas | Tabula Muris |
|---|---|---|---|
| MAGIC | 0.18866 +/- 0.00002 | 0.23186 +/- 0.00003 | 0.18417 +/- 0.00001 |
| DCA | 0.21843 +/- 0.00029 | 0.26834 +/- 0.00033 | 0.21595 +/- 0.00066 |
| ALRA | 0.28233 +/- 0.00074 | 0.33169 +/- 0.00004 | 0.28000 +/- 0.00061 |
| *scDiff* | 0.18102 +/- 0.00109 | 0.20794 +/- 0.00265 | 0.16800 +/- 0.00227 |

To benchmark scDiff with existing leaderboards, we present results following the setting of Open-Problems in denoising and cell type annotation. For the annotation task, we copied the accuracy of four top-performing baselines from the leaderboard with random splits. The results are summarized in Table 11, where scDiff is evaluated across five runs. We observe that scDiff matches the best performance of the leaderboard.

For the denoising task, we selected three performant baselines from the leader board regarding MSE. We report the mean and variance of scDiff and baselines across five runs in Table 12. Note that the MSE results on leader board are scaled for better illustration, we reproduced the benchmark for a fair comparison via the code repository of OpenProblems without scaling. A conclusion can be drawn that scDiff outperforms the baselines across datasets.

