# OpenReview forum: "A General Single-Cell Analysis Framework via Conditional Diffusion Generative Models"
_ICLR.cc/2024/Conference — Submitted to ICLR 2024_

### Official Review · Reviewer_Abad · 2023-10-28

**Soundness:** 2 fair
**Presentation:** 2 fair
**Contribution:** 2 fair
**Rating:** 6
**Confidence:** 4

**Summary:**

The paper tries to form three common single-cell analysis, cell labeling, expression completion and knowledge transfer, into a unified posterior estimation problem by leveraging the ability of learning data distribution of diffusion model. The idea is general reasonable and proved effective compared with baselines, but the paper still needs to be further clarified and improved.

**Strengths:**

(1) The problem that the paper tries to address is significant in the field of analyzing single-cell data.
(2) The idea of forming three major tasks of analyzing single cell data into posterior distribution is doable.
(3) The paper illustrate the method clear in its Figure 1.
(4) The datasets employed to evaluate the method is comprehensive.

**Weaknesses:**

(1) The paper mentions using LLMs as the prior, but why not compare the proposed method with LLMs-based ones, such as those in [1]?
(2) From the part of the paper under Section 2.2 to equation (9), the author used a large amount of words to introduce basics of the diffusion model. I would suggest the author to move this part to appendix as this is not the contribution of this paper.
(3) For the conditioner part, how does the cross-attention exactly used? Will it be able to automatically detect which task the model is focusing on?
(4) There are quite a few confusing part in the paper. For instance, the paper claims that it forms the task into a posterior estimation, but how the prior is used in the model? I don't think the paper explicitly explains this. Also, for cell labeling and knowledge transfer, I don't thinks it's a generation task so that generative model is a good practice in this case. Intuitively cell lebeling and knowledge transfer is more like a prediction task to me.
(5) I don't feel the strong motivation of using a attention mechanism to help the model learn specific tasks. To me it's more like the proposed method assembles three tasks very hard to let the model to learn. Therefore, the proposed method lacks novelty as each individual task can still be solved in the same framework.
(6) I'm also confused about the current formation of the x_\theta(x_0, \epsilon, c, t). Why not just follow the standard way as in CV that just uses U-Net? I strongly suggest the author to clarify the motivation of their model structure.

[1] Liu et al., Evaluating the Utilities of Large Language Models in Single-cell Data Analysis

**Questions:**

Please refer to my comments in "Weakness".

**Details Of Ethics Concerns:**

The paper uses single-cell gene expression data, and it needs ethical statement in the paper.

---

> ### Author Response · Authors · 2023-11-17
> **Reply to Reviewer Abad #1**
>
> > W1. The paper mentions using LLMs as the prior, but why not compare the proposed method with LLMs-based ones, such as those in [1]?
> >
> >
> > [1] Liu et al., Evaluating the Utilities of Large Language Models in Single-cell Data Analysis
> >
>
> A. We thank the reviewer for the suggestion. We first note that **reference [1] does not consider LLMs that handle natural languages**. Instead, the LLM it referred to are large-scale pre-trained models using single-cell omics data, such as scGPT [2], scBERT [3], and Geneformer [4].
>
> Nevertheless, to provide more detailed comparisons, we ran scGPT, and Geneformer on our annotation benchmarking datasets. The annotation accuracies across three runs are summarized in the following table. We excluded scBERT due to its fine-tuning time exceeding 24 hours. Our results indicate that scDiff performs comparably against these large-scale pre-trained models, even without pre-training on hundreds of thousands of single-cell samples.
>
> | ACC | PBMC12K | Pancreas | HLCA | Immune | Brain | Liver |
> | --- | --- | --- | --- | --- | --- | --- |
> | Geneformer | 0.9787 +/- 0.0019 | -- (no ENSG) | 0.8328 +/- 0.0057 | 0.8556 +/- 0.0098 | 0.9338 +/- 0.0071 | 0.8711 +/- 0.0390 |
> | scGPT | 0.9630 +/- 0.0055 | 0.9454 +/- 0.0374 | 0.8634 +/- 0.0200 | 0.9071 +/- 0.0108 | 0.9504 +/- 0.0029 | 0.8640 +/- 0.0379 |
> | scDiff | 0.9670 +/- 0.0042 | 0.9680 +/- 0.0143 | 0.8931 +/- 0.0070 | 0.8442 +/- 0.0076 | 0.9473 +/- 0.0074 | 0.8439 +/- 0.0042 |
>
> [2] Cui, Haotian, et al. "scgpt: Towards building a foundation model for single-cell multi-omics using generative ai." bioRxiv (2023): 2023-04.
>
> [3] Yang, Fan, et al. "scBERT as a large-scale pretrained deep language model for cell type annotation of single-cell RNA-seq data." Nature Machine Intelligence 4.10 (2022): 852-866.
>
> [4] Theodoris, Christina V., et al. "Transfer learning enables predictions in network biology." Nature (2023): 1-9.
>
> > W2. From the part of the paper under Section 2.2 to equation (9), the author used a large amount of words to introduce basics of the diffusion model. I would suggest the author to move this part to appendix as this is not the contribution of this paper.
> >
>
> A. We thank the reviewer for the suggestion. Indeed, the formulation of DDPM is not our contribution. However, this brief introduction provides the necessary background and the notations for our method, making it indispensable. We understand the reviewer’s concern about the clarity of our contribution. Thus, we have adjusted the manuscript to explicitly indicate that the DDPM introduction is preliminary to our method.
>
> > W3. For the conditioner part, how does the cross-attention exactly used? Will it be able to automatically detect which task the model is focusing on?
> >
>
> A. We thank the reviewer for the question and the opportunity for us to clarify this matter. Cross-attention combines **the cell-embedding with the condition information**, e.g., cell type, cell perturbation, and gene perturbation. Specifically, the query (Q) for each sample is the cell embedding, and the key (K) and value (V) are derived from the condition representations.
>
> The cross-attention module is **downstream task-agnostic**, meaning it will function exactly the same regardless of the intended downstream task. In fact, the training objective of scDiff is already task-agnostic (the primary highlight of scDiff): it is trained to *recover clean gene expression patterns from noised ones, given the condition information associated with the cell*.
>
> *To be continued in the next reply*

---

> ### Author Response · Authors · 2023-11-17
> **Reply to Reviewer Abad #2**
>
> > W4. There are quite a few confusing part in the paper. For instance, the paper claims that it forms the task into a posterior estimation, but how the prior is used in the model? I don't think the paper explicitly explains this. Also, for cell labeling and knowledge transfer, I don't thinks it's a generation task so that generative model is a good practice in this case. Intuitively cell lebeling and knowledge transfer is more like a prediction task to me.
> >
>
> A. We thank the reviewer for the questions regarding the prior and the formulation of the annotation task as posterior generation. First, the **priors are the conditions used in the scDiff model**, e.g., cell types, perturbation, and gene perturbation. They are essential for realizing the downstream tasks. For example, (1) annotation is implemented by choosing the condition that results in the lowest reconstruction error for a given input cell, and (2) perturbation is implemented by applying the perturbed condition to the normal (unperturbed) cell.
>
> Second, regarding the cell annotation task formulation as posterior estimation, we understand that the typical approach to classification is by using a “deterministic prediction head”. However, we point out that the goal of cell labeling and knowledge transfer can indeed be formulated as *estimating the posterior of gene expression under certain cell types or unknown conditions*, which is a generative task. We further note that **formulating classification problems into generative task is not uncommon** in recent machine learning communities. A notable example is using autoregressive generative language models to solve multiple choice (multi-class classification) or answer True/False (binary classification).
>
> > W5. I don't feel the strong motivation of using a attention mechanism to help the model learn specific tasks. To me it's more like the proposed method assembles three tasks very hard to let the model to learn. Therefore, the proposed method lacks novelty as each individual task can still be solved in the same framework.
> >
>
> A. We thank the reviewer for the series of questions regarding cross-attention usage in scDiff. First, we clarify that we *do not assemble tasks* using the cross-attention. Instead, as the highlight of our method, scDiff is trained in a *task-agnostic* manner by reconstructing clean gene expression from the noised one, given its corresponding conditions (e.g., cell type, perturbation, and gene perturbation). Similar to latent diffusion [5], the cross-attention here servers to fuse prior (condition) information into the model while not explicitly selecting one of them.
>
> Second, we use cross-attention to combine the cell information with the condition information due to its **effectiveness and superior performance** in the downstream evaluations. A more naive implementation to combine cell embedding and condition information is directly adding up their embeddings (followed by a feed-forward network), rather than fusing through a cross-attention mechanism. To demonstrate the superiority of cross-attention over the naive adding approach, we included an additional ablation study comparing the two for the annotation tasks. We observe that cross-attention resulted in higher accuracy than the naive adding counterpart, with the only exceptions being Liver and Pancreas.
>
> |  | Brain | HLCA | Immune | Liver | Pancreas | PBMC12K |
> | --- | --- | --- | --- | --- | --- | --- |
> | Default | 0.9473 +/- 0.0074 | 0.8931 +/- 0.0070 | 0.8442 +/- 0.0076 | 0.8439 +/- 0.0042 | 0.9680 +/- 0.0143 | 0.9670 +/- 0.0042 |
> | No cross-attention | 0.9376 +/- 0.0019 | 0.8870 +/- 0.0012 | 0.8318 +/- 0.0028 | 0.8456 +/- 0.0054 | 0.9722 +/- 0.0014 | 0.9608 +/- 0.0042 |
>
> [5] Rombach, Robin, et al. "High-resolution image synthesis with latent diffusion models." *Proceedings of the IEEE/CVF conference on computer vision and pattern recognition*. 2022.
>
> *To be continued in the next reply.*

---

> > ### Comment · Reviewer_Abad · 2023-11-21
> > **Response to author**
> >
> > Thank the author for the comprehensive response. My concerns has been addressed.

---

> > > ### Author Response · Authors · 2023-11-22
> > > **Thanks for your response**
> > >
> > > Thanks for your response and support. We are glad to know that our rebuttal has addressed your concerns. Please let us know in case there still remain outstanding concerns, and if so, we will be happy to respond.

---

> ### Author Response · Authors · 2023-11-17
> **Reply to Reviewer Abad #3**
>
> > W6. I'm also confused about the current formation of the $x_\theta(x_0, \epsilon, c, t)$. Why not just follow the standard way as in CV that just uses U-Net? I strongly suggest the author to clarify the motivation of their model structure.
> >
>
> A. We appreciate the question raised by the reviewer. We divide questions into three parts and answer them in detail in the following paragraphs. We summarize the replies as (i) further explain the components of scDiff, (ii) traditional U-Net with 2D convolution is not applicable to our case, (iii) we have attempted our best to integrate ideas from U-Net and demonstrate the effectiveness of our original design, and (iv) elaborate on the motivation of *scDIff* architecture.
>
> First, we briefly expand the intuition of our model architecture. Our formulation of the scDiff model comprises four major parts: (1) the *embedder* $\phi$ that turns the cell’s gene expression profile into an embedding; (2) the *conditioners* $\psi_f$ that extracts condition embeddings from the given cell’s conditions; (3) the *encoder* $\mathcal{E}$ that fuses information of the cell embedding and the condition embeddings; and finally, (4) the *decoder* $\mathcal{D}$ that projects the processed cell embedding back to the gene expression space for reconstruction. This formulation can be seen as a variation to the standard feed-forward multi-layer perceptron (MLP), where the cross-attention mechanism only serves to combine the cell embedding and condition embeddings.
>
> Second, we did not directly apply a standard U-Net with 2D convolution in our case due to the distinct characteristic of single-cell data compared to image data, requiring us to make necessary adaptations and redesign to the model architecture. To begin with, we note that **2D convolution is not applicable in our case** since *single-cell RNAseq data is tabular*: there is no canonical ordering in the pixel space (gene transcripts). Conversely, the pixels in images have a canonical ordering from their spatial coordinate, making it viable to apply 2D convolution to leverage the inductive bias in images. In the single-cell domain, simple MLP-style architecture has shown effectiveness with promising performance. **MLP is thus widely used as the backbone in many single-cell deep learning applications**, including the strong baselines we included, such as scANVI, DCA, and scGEN. Thus, we learned from these prior works and settled on using an MLP backbone, with the addition of cross-attention to fuse information from cell embedding and condition embeddings.
>
> Third, **we have attempted our best to integrate ideas from the standard U-Net into our scDiff model**. (1) Our *scDiff mimics the residual connections in U-Net*. Particularly, as detailed in the provided Appendix C.2, we explained and discussed the design of the *context conditioner* with a *reversed mixing with the cell embeddings*. This reversed mixing - first layer cell embedding is mixed with last layer context conditioner embedding, and vice versa - is inspired by the recently proposed DiffMAE [6], which also discussed the relation of this design with U-Net’s residual connection. (2) Going one step beyond, we have carried out an additional ablation study that more directly investigates the *effect of the U-Net residual* component by adding residual connections between the first and last encoder layers, etc. We observe that adding the U-Net residual component results in minor improvements to the performance for 3/6 annotation datasets. This additional result indicates the effectiveness of our original design, which is simpler, and also highlights the potential of scDiff architecture to be further optimized in the future to achieve even better performance.
>
> |  | Brain | HLCA | Immune | Liver | Pancreas | PBMC12K |
> | --- | --- | --- | --- | --- | --- | --- |
> | Default | 0.9473 +/- 0.0074 | 0.8931 +/- 0.0070 | 0.8442 +/- 0.0076 | 0.8439 +/- 0.0042 | 0.9680 +/- 0.0143 | 0.9670 +/- 0.0042 |
> | U-Net residual | 0.9456 +/- 0.0047 | 0.8959 +/- 0.0050 | 0.8502 +/- 0.0075 | 0.8399 +/- 0.0103 | 0.9693 +/- 0.0014 | 0.9644 +/- 0.0021 |
>
> *To be continued in the next reply*

---

> ### Author Response · Authors · 2023-11-17
> **Reply to Reviewer Abad #4**
>
> Finally, we explain our motivation of current model architecture accompanied with ablation studies that systematically tested the effectiveness of various parts of our scDiff model. We present the ablation studies in annotation task regarding number of layers in embedder and decoder, adding residual connection and removing cross-attention. The results are summarized in the following table, where we report the accuracy on all the annotation datasets across 5 runs. The autoencoder structure has shown great success in single-cell analysis [7, 8]. Following the existing works [9,10], we build our model with a relatively heavy 6-layer encoder and a light 1-layer decoder. Before the encoder, we utilize an embedder to project the input expression into the embedding space. In the encoder, we wish to inject the prior information through different conditions into the expression embeddings, where we choose the cross-attention mechanism due to performance concerns. We extract the embeddings of conditions through various conditioners and feed them into the key and value of the cross-attention blocks. The embedder serves as a projection from expression space to embedding space, while the decoder reverses this process. We empirically observed that a 1-layer neural network is sufficient for such transformation.
>
> |  |  | Brain | HLCA | Immune | Liver | Pancreas | PBMC12K |
> | --- | --- | --- | --- | --- | --- | --- | --- |
> |  | Default  | 0.9473 +/- 0.0074 | 0.8931 +/- 0.0070 | 0.8442 +/- 0.0076 | 0.8439 +/- 0.0042 | 0.9680 +/- 0.0143 | 0.9670 +/- 0.0042 |
> | Decoder | 2-layer | 0.9445 +/- 0.0041 | 0.8911 +/- 0.0042 | 0.8444 +/- 0.0070 | 0.8589 +/- 0.0238 | 0.9735 +/- 0.0023 | 0.9700 +/- 0.0045 |
> |  | 3-layer | 0.9378 +/- 0.0017 | 0.8994 +/- 0.0038 | 0.8383 +/- 0.0112 | 0.8062 +/- 0.0216 | 0.9731 +/- 0.0014 | 0.9701 +/- 0.0029 |
> |  | 4-layer | 0.9409 +/- 0.0064 | 0.8989 +/- 0.0037 | 0.8414 +/- 0.0051 | 0.8199 +/- 0.0185 | 0.9763 +/- 0.0011 | 0.9744 +/- 0.0002 |
> | Embedder | 2-layer | 0.9394 +/- 0.0029 | 0.8857 +/- 0.0089 | 0.8424 +/- 0.0061 | 0.8434 +/- 0.0239 | 0.9720 +/- 0.0011 | 0.9667 +/- 0.0039 |
> |  | 3-layer | 0.9512 +/- 0.0048 | 0.8847 +/- 0.0053 | 0.8358 +/- 0.0081 | 0.8447 +/- 0.0224 | 0.9745 +/- 0.0026 | 0.9672 +/- 0.0039 |
> |  | 4-layer | 0.9434 +/- 0.0048 | 0.8927 +/- 0.0018 | 0.8461 +/- 0.0014 | 0.8423 +/- 0.0100 | 0.9742 +/- 0.0012 | 0.9606 +/- 0.0036 |
> | General | No cross-attention | 0.9376 +/- 0.0019 | 0.8870 +/- 0.0012 | 0.8318 +/- 0.0028 | 0.8456 +/- 0.0054 | 0.9722 +/- 0.0014 | 0.9608 +/- 0.0042 |
> |  | Remove diffusion | 0.5513 +/- 0.1390 | 0.1487 +/- 0.0313 | 0.1233 +/- 0.1661 | 0.1525 +/- 0.0214 | 0.3749 +/- 0.0627 | 0.5939 +/- 0.1144 |
>
> [6] Wei, Chen, et al. "Diffusion Models as Masked Autoencoders." *arXiv preprint arXiv:2304.03283* (2023).
>
> [7] Lopez, Romain, et al. "Deep generative modeling for single-cell transcriptomics." *Nature methods* 15.12 (2018): 1053-1058.
>
> [8] Eraslan, Gökcen, et al. "Single-cell RNA-seq denoising using a deep count autoencoder." *Nature communications* 10.1 (2019): 390.
>
> [9] Gong, Jing, et al. "xTrimoGene: An Efficient and Scalable Representation Learner for Single-Cell RNA-Seq Data." *NeurIPS* (2023).
>
> [10] He, Kaiming, et al. "Masked autoencoders are scalable vision learners." *Proceedings of the IEEE/CVF conference on computer vision and pattern recognition*. 2022.
>
> > W7. The paper uses single-cell gene expression data, and it needs ethical statement in the paper. Privacy, security and safety; Responsible research practice (e.g., human subjects, data release).
> >
>
> A. We thank the reviewer for carefully pointing this out. We note that all data used are from public resources and thus adhere to their license agreements. We can confirm that no known privacy and ethics issues are associated with the public data we chose to use. We have also added the ethics statement to clarify the above, as suggested by the reviewer.

---

### Official Review · Reviewer_Z48j · 2023-11-01

**Soundness:** 2 fair
**Presentation:** 3 good
**Contribution:** 2 fair
**Rating:** 6
**Confidence:** 3

**Summary:**

The paper introduces a novel single-cell analysis framework, scDiff, which approaches various computational tasks through posterior approximation. scDiff comprises three key components: a conditional diffusion model for posterior approximation, a series of encoders that enocod cell conditions into numerical vectors, and a cross-attention unit that combines these condition embeddings. scDiff also has an adaptable structure, allowing the incorporation of text as prior knowledge.

To evaluate the model's performance, the authors conducted experiments covering a range of benchmarking tasks. These tasks included cell annotation, prediction of missing values, identification of novel cell types, and annotating cells using just one or a few samples. The results demonstrated that scDiff achieves competitive performance when compared to state-of-the-art models across multiple datasets and task categories.

**Strengths:**

The authors tackle a range impactful problems in the field of single-cell analysis, including tasks such as cell type annotation, imputation, novel type identification, and perturbation prediction for scRNA-seq data. Typically, each of these tasks would demand a separate model.

Despite the existence of extensive literature on each of these individual problems, the authors suggested a unifying framework that encompasses multiple benchmark tasks within a single, cohesive framework.

**Weaknesses:**

**Major:**

- *Novelty:* It is not clear what the novel aspect of scDiff model is from a machine learning perspective. Although the authors introduced posterior inference as a novel unified framework for several single-cell tasks, posterior inference through variational or generative processes is a well-explored area, even within the field of single-cell analysis.

- *Contributions:* The reported results in the experimental sections do not convincingly demonstrate that scDiff significantly outperforms relevant models for the specific tasks at hand. Furthermore, the authors have not quantified the computational cost associated with having a unified model that covers multiple tasks, particularly in relation to the observed improvements. While the enhanced performance might not necessarily stem from the model's extension, it could be attributed to factors like the use of the diffusion process, attention units, or different implementations.


**Minor:**

- Some of the experimental settings descriptions, such as those outlined in section 4.2.1, are not clearly explained and can be rather confusing. Additional clarification is needed in this regard.

- In Table 1, it would be valuable to include the number of clusters and chance level for each dataset to provide a more comprehensive understanding of the results.

**Questions:**

- In the context of the missing value imputation task, the assumption that all zero-expressed genes are missing and no actual zero-expressed genes exist may not be entirely accurate. It might be more biologically relevant for the model to learn the mask matrix, $M$, instead of assuming that $x_g > 0$. In biology, we know that non or less-expressed genes can still play a marker role in some cell types.

- Can you provide further elaboration on Equation 10? It is not entirely clear why the single-cell data are encoded using the suggested *"TimeEmbed"* function.

- Why does scDiff utilize a linear encoder / decoder?

- It would be insightful to understand the computational efficiency of using a model like scDiff compared to an equivalent model designed solely for solving one or a few downstream tasks.

- In Table 2a, for the missing value imputation task, the authors reported correlation values. However, is not the primary goal to approximate the value of gene expression rather than capturing the overall expression pattern (correlation at the gene population level)? It might be more informative if the authors report the average (normalized) error.

- In Table 2, it would be beneficial to include the number of genes used for each task per dataset.

- For the study in Table 2a, is there any consideration for zero-expressed genes?

- What is the chance level in Figure 2? Is not the number of cell types limited in this context?

- In Figure 3, why the top and bottom subfigures do not reveal the same relative performance pattern? Could you provide further elaboration on how they are related to each other?

---

> ### Author Response · Authors · 2023-11-17
> **Reply to Reviewer Z48j #1**
>
> > W1. *Novelty:* It is not clear what the novel aspect of scDiff model is from a machine learning perspective. Although the authors introduced posterior inference as a novel unified framework for several single-cell tasks, posterior inference through variational or generative processes is a well-explored area, even within the field of single-cell analysis.
> >
>
> A. Thank you for raising the concern about novelty. There are some existing works utilizing variational or generative processes in single-cell analysis [1, 2], but they are limited to specific type of tasks. To the best of our knowledge, we are the first to present the idea of handling multiple types of tasks (e.g., cell type annotation, imputation and perturbation prediction) without modifying the model structure. Our approach is novel, utilizing conditional diffusion models with prior information. We empirically find out that with task-agnostic training process and task-specific generative process, *scDiff* is able to match or even outperform the current SOTA. It is important to note that the modularized conditioning strategy can be readily extended to further tasks, as long as their objectives can be formulated as or approximated by the posterior.
>
> [1] Lopez, Romain, et al. "Deep generative modeling for single-cell transcriptomics." *Nature methods* 15.12 (2018): 1053-1058.
>
> [2] Lotfollahi, Mohammad, F. Alexander Wolf, and Fabian J. Theis. "scGen predicts single-cell perturbation responses." *Nature methods* 16.8 (2019): 715-721.
>
> > W2. *Contributions:* The reported results in the experimental sections do not convincingly demonstrate that scDiff significantly outperforms relevant models for the specific tasks at hand. Furthermore, the authors have not quantified the computational cost associated with having a unified model that covers multiple tasks, particularly in relation to the observed improvements. While the enhanced performance might not necessarily stem from the model's extension, it could be attributed to factors like the use of the diffusion process, attention units, or different implementations.
> >
>
> A. We thank the reviewer for pointing out the concerns. We formulate our reponses in three aspects: (i) we justify the performance of *scDiff* and provide additional benchmark results, (ii) *scDiff* has a computational complexity that is nearly the same as a standard MLP, and (iii) we validate our design of *scDiff* through ablation studies.
>
> We first point out that scDiff does show **convincingly superior performance**: it achieved top performance for 11 out of the 15 primary benchmarks we have carried out. For the remaining four tasks, scDiff performed comparably to the best methods. More importantly, scDiff achieved top performances in diverse tasks with a single **unified framework**, unlike other baselines that are task-specific. Our wide variety of conditioners, spanning from the simple class embedding, to LLM, to graph neural networks, also highlight the **extendability** of our method as another significant novelty.
>
> In addition, to benchmark *scDiff* with existing leaderboards, we present results follow the setting of OpenProblems in [denoising](https://openproblems.bio/results/denoising/) and [cell type annotation](https://openproblems.bio/results/label_projection/). For annotation task, we copied the accuracy of four top-performing baselines from the [leaderboard](https://openproblems.bio/results/label_projection/) with random splits. The results are as follows, where *scDiff* is evaluated across five runs. We observe that *scDiff* matches the best performance of the leaderboard.
>
> | ACC | Pancreas | Tabula Muris | CeNGEN |
> | --- | --- | --- | --- |
> | Logistic regression (log CP10k) | 0.98 | 0.92 | 0.89 |
> | Seurat reference mapping (SCTransform) | 0.98 | 0.90 | 0.83 |
> | Multilayer perceptron (log scran) | 0.98 | 0.92 | 0.87 |
> | XGBoost (log CP10k) | 0.97 | 0.86 | 0.84 |
> | scDiff | 0.9800 +/- 0.0039 | 0.9257 +/- 0.0066 | 0.8889 +/- 0.0024 |
>
> For denoising task, we selected three performant baselines from the [leaderboard](https://openproblems.bio/results/denoising/) regarding MSE. We report the mean and variance of *scDiff* and baselines across five runs. Note that the MSE results on [leaderboard](https://openproblems.bio/results/denoising/) are scaled for better illustration, we reproduced the benchmark for a fair comparison via the [public code repository](https://github.com/openproblems-bio/openproblems/tree/main/openproblems/tasks/denoising) without scaling. A conclusion can be drawn that *scDiff* outperforms the baselines across datasets.
>
> | MSE | PBMC | Pancreas | Tabula Muris |
> | --- | --- | --- | --- |
> | MAGIC | 0.1887 +/- 0.0001 | 0.2319 +/- 0.0001 | 0.1842 +/- 0.0001 |
> | DCA | 0.2184 +/- 0.0003 | 0.2683 +/- 0.0003 | 0.2160 +/- 0.0007 |
> | ALRA | 0.2823 +/- 0.0007 | 0.3317 +/- 0.0001 | 0.2800 +/- 0.0006 |
> | scDiff | 0.1810 +/- 0.0011 | 0.2079 +/- 0.0027 | 0.1680 +/- 0.0023 |
>
> *To be continued in the next reply.*

---

> ### Author Response · Authors · 2023-11-17
> **Reply to Reviewer Z48j #2**
>
> Regarding the computational complexity, we note that ***scDiff* has a computational complexity that is nearly the same as a standard MLP**. In the following table, we report the per-epoch computation time for scDiff against standard baselines for all our benchmarking tasks. It is crucial to acknowledge that diffusion models inherently demand a relatively longer time during the inference stage compared to other generative models. In our empirical observations, we find that the inference time of *scDiff* is negligible compared to the training time. Therefore, our emphasis is placed on training speed.
>
> From the table, we observe that in all cases, scDiff has comparable per-epoch runtime as the baselines, especially for MLP-based ones (scANVI, DCA, and scGEN). Referring to the model structure of *scDiff*, the central backbone of it comprises the cross-attention (CA) encoder blocks, where the tokens are the number of types of conditions (e.g., cell type, perturbation status). As mentioned in Appendix E.1 Table 3, the number of types of conditions will not exceed $5$. Thus, unlike the traditional transformer model that poses $O(N^2)$ complexity, where the token number $N$ is typically much larger than the hidden dimensions, in *scDiff* the feedforward module dominates the computation complexity of our CA block.
>
> For the space complexity, we note that the required amount of GPU memory is determined by both the batch size and the number of genes. Empirically, we set $\text{batch size} = 2048$ throughout all the experiments. On the dataset with the highest number of genes, namely HLCA with 28,024 genes, *scDiff* necessitates 4.5 GB of GPU memory. Given the above discussion on training efficiency, it is noteworthy that *scDiff* can be scaled effectively to large datasets.
>
> |  | Dataset | scDiff per-epoch runtime (sec) | Baseline per-epoch runtime (sec) | Baseline name |
> | --- | --- | --- | --- | --- |
> | Annotation | Brain | 9.63 | 9.82 | scANVI |
> |  | HLCA | 13.60 | 12.73 | scANVI |
> |  | Immune | 5.28 | 8.86 | scANVI |
> |  | Liver | 2.83 | 2.97 | scANVI |
> |  | Pancreas | 5.05 | 14.37 | scANVI |
> |  | PBMC12K | 2.04 | 3.31 | scANVI |
> | Denoising | 293T | 0.8847 | 0.53 | DCA |
> |  | Jurkat | 0.5736 | 0.60 | DCA |
> |  | PBMC1K | 0.2634 | 0.16 | DCA |
> | Perturbation | HPoly | 1.96 | 1.83 | scGEN |
> |  | PBMC | 3.16 | 3.53 | scGEN |
> |  | Salmonella | 1.58 | 1.81 | scGEN |
> | Gene Perturbation | Adamson | 83.88 | 1189.66 | GEARS |
> |  | Dixit | 44.67 | 760.70 | GEARS |
> |  | Norman | 155.88 | 491.97 | GEARS |
>
> Finally, we have carried out additional experiments to **systematically test the effectiveness of scDiff**, including the usage of the diffusion process, cross-attention, and embedder/decoder types. Our results highlight the importance of the diffusion process, while other minor architectural designs, such as embedder and decoder, show a less noticeable impact on the final performance.
>
> |  |  | Brain | HLCA | Immune | Liver | Pancreas | PBMC12K |
> | --- | --- | --- | --- | --- | --- | --- | --- |
> |  | Default | 0.9473 +/- 0.0074 | 0.8931 +/- 0.0070 | 0.8442 +/- 0.0076 | 0.8439 +/- 0.0042 | 0.9680 +/- 0.0143 | 0.9670 +/- 0.0042 |
> | Decoder | 2-layer | 0.9445 +/- 0.0041 | 0.8911 +/- 0.0042 | 0.8444 +/- 0.0070 | 0.8589 +/- 0.0238 | 0.9735 +/- 0.0023 | 0.9700 +/- 0.0045 |
> |  | 3-layer | 0.9378 +/- 0.0017 | 0.8994 +/- 0.0038 | 0.8383 +/- 0.0112 | 0.8062 +/- 0.0216 | 0.9731 +/- 0.0014 | 0.9701 +/- 0.0029 |
> |  | 4-layer | 0.9409 +/- 0.0064 | 0.8989 +/- 0.0037 | 0.8414 +/- 0.0051 | 0.8199 +/- 0.0185 | 0.9763 +/- 0.0011 | 0.9744 +/- 0.0002 |
> | Embedder | 2-layer | 0.9394 +/- 0.0029 | 0.8857 +/- 0.0089 | 0.8424 +/- 0.0061 | 0.8434 +/- 0.0239 | 0.9720 +/- 0.0011 | 0.9667 +/- 0.0039 |
> |  | 3-layer | 0.9512 +/- 0.0048 | 0.8847 +/- 0.0053 | 0.8358 +/- 0.0081 | 0.8447 +/- 0.0224 | 0.9745 +/- 0.0026 | 0.9672 +/- 0.0039 |
> |  | 4-layer | 0.9434 +/- 0.0048 | 0.8927 +/- 0.0018 | 0.8461 +/- 0.0014 | 0.8423 +/- 0.0100 | 0.9742 +/- 0.0012 | 0.9606 +/- 0.0036 |
> | General | No cross-attention | 0.9376 +/- 0.0019 | 0.8870 +/- 0.0012 | 0.8318 +/- 0.0028 | 0.8456 +/- 0.0054 | 0.9722 +/- 0.0014 | 0.9608 +/- 0.0042 |
> |  | U-Net residual | 0.9456 +/- 0.0047 | 0.8959 +/- 0.0050 | 0.8502 +/- 0.0075 | 0.8399 +/- 0.0103 | 0.9693 +/- 0.0014 | 0.9644 +/- 0.0021 |
> |  | Remove diffusion | 0.5513 +/- 0.1390 | 0.1487 +/- 0.0313 | 0.1233 +/- 0.1661 | 0.1525 +/- 0.0214 | 0.3749 +/- 0.0627 | 0.5939 +/- 0.1144 |
>
> *To be continued in the next reply*

---

> ### Author Response · Authors · 2023-11-17
> **Reply to Reviewer Z48j #3**
>
> > W3. Some of the experimental settings descriptions, such as those outlined in section 4.2.1, are not clearly explained and can be rather confusing. Additional clarification is needed in this regard.
> >
>
> A. Thank you for suggestion of improving the clarity of our experiments’ descriptions. We have made according changes in Appendix E. Here we summarize the experimental setting of Section 4.2.1: For a specific dataset, we refer to the cell types that contain more cells than a threshold as majority types and the remaining as minority types. We randomly sample one cell per type from the minority types as the one-shot set, and the rest of the cells in minority types are used for evaluation. *scDiff* is first pre-trained on all cells in majority types and then fine-tuned on the one-shot set for $50$ epochs.
>
> > W4. In Table 1, it would be valuable to include the number of clusters and chance level for each dataset to provide a more comprehensive understanding of the results.
> >
>
> A. We have provided the number of clusters (cell types) in our appendix (see Appendix Table 4). We decided to defer the dataset statistics, such as the number of cell types to the appendix to make the main presentations more compact. Nevertheless, to better guide readers find this information, we have adjusted the main text to explicitly refer the readers to the appendix for dataset statistics.
>
> Furthermore, we have added the chance level by predicting the majority class all the time to the annotation results table to provide better sense of the performance as suggested.
>
> > Q1. In the context of the missing value imputation task, the assumption that all zero-expressed genes are missing and no actual zero-expressed genes exist may not be entirely accurate. It might be more biologically relevant for the model to learn the mask matrix, M, instead of assuming that $x_g > 0$. In biology, we know that non or less-expressed genes can still play a marker role in some cell types.
> >
>
> A. We thank the reviewer for the question regarding our benchmarking strategy for the denoising task. We clarify that we do not assume all zero-expressed genes are missing. Specifically, we randomly mask out 10% of the non-zero entries and turn them into zeros in the training data. The assumption here is that all zero-expressed genes are "true zeros," following the benchmarking setting from a well-established paper [1].
>
> Meanwhile, we understand the reviewer's concern about the simplistic assumption about zero-expressed genes. Thus, we have carried out an additional set of denoising benchmarks using molecular cross-validation (MCV) [2] to comprehensively evaluate our proposed method's biological relevance. MCV is a more rigorous data-splitting strategy that partitions the molecules in a cell into two parts. According to the experimental results of the newly added OpenProblems benchmarks in the replies to W2, we showed that scDiff outperforms the top performance compared to the baseline approaches.
>
> [1] Van Dijk, David, et al. "Recovering gene interactions from single-cell data using data diffusion." *Cell* 174.3 (2018): 716-729.
>
> [2] Batson, Joshua, Loïc Royer, and James Webber. "Molecular cross-validation for single-cell RNA-seq." *BioRxiv* (2019): 786269.
>
> > Q2. Can you provide further elaboration on Equation 10? It is not entirely clear why the single-cell data are encoded using the suggested *"TimeEmbed"* function.
> >
>
> A. In a nut shell, the *TimeEmbed* function is used to encode the noise level of the diffusion process. Diffusion denoising probablistic models (DDPM) [1] works by gradually denoising a signal starting from a normal Gaussian noise. Informally, the TimeEmbed function serves to inform the model at which stage the denoising process is at, and whether the current input signal is expected to be more or less noisy. We follow this TimeEmbed strategy from DDPM by adding the time embedding to the raw expression embedding.
>
> [1] Ho, Jonathan, Ajay Jain, and Pieter Abbeel. "Denoising diffusion probabilistic models." *Advances in neural information processing systems* 33 (2020): 6840-6851.
>
> > Q3. Why does scDiff utilize a linear encoder / decoder?
> >
>
> A. We use linear Embedder and Decoder for simplicity. Conversely, the Encoder is implemented as cross-attention blocks that integrate the encoded cell embedding with the condition information. To better illustrate the effectiveness of linear embedder and decoder, we have included a new ablation study that replaces linear embedder/decoder with multi-layer perceptron (MLP) counterparts (see ablation study results in the response to W2). Our results show that although MLP embedder/decoder improves the performance over the linear counterparts in some datasets, their performance is comparable overall, indicating the effectiveness of our original design choice.
>
> *To be continued in the next reply*

---

> ### Author Response · Authors · 2023-11-17
> **Reply to Reviewer Z48j #4**
>
> > Q4. It would be insightful to understand the computational efficiency of using a model like scDiff compared to an equivalent model designed solely for solving one or a few downstream tasks.
> >
>
> A. We thank the reviewer for suggesting analyzing the computational cost of scDiff against standard baselines. We note that **scDiff has a computational complexity that is nearly the same as a standard MLP.** The results are summarized in the reply to W2. We briefly explain this claim in the following and show empirical results supporting it.
>
> The central backbone of scDiff comprises the cross-attention (CA) encoder blocks, where the tokens are the number of conditions (e.g., cell type, cell perturbation, gene perturbation). Thus, unlike the traditional transformer model that poses $O(N^2)$ complexity, where the token number $N$ is typically much larger than the hidden dimensions, the feedforward module dominates the computation complexity of our CA block.
>
> > Q5. In Table 2a, for the missing value imputation task, the authors reported correlation values. However, is not the primary goal to approximate the value of gene expression rather than capturing the overall expression pattern (correlation at the gene population level)? It might be more informative if the authors report the average (normalized) error.
> >
>
> A. As we alluded to in our response to Q1, we follow the benchmarking strategy described by the original paper [1], which suggested Pearson correlation for evaluation. Nonetheless, in the newly added MCV experiments, we followed OpenProblems and reported MSE metrics. In either case, scDiff shows comparable or better performance than the baselines.
>
> [1] Van Dijk, David, et al. "Recovering gene interactions from single-cell data using data diffusion." *Cell* 174.3 (2018): 716-729.
>
> > Q6. In Table 2, it would be beneficial to include the number of genes used for each task per dataset.
> >
>
> A. We thank the reviewer for the suggestion. We have already provided basic dataset statistics in the Appendix (see Table 4), including the number of genes, batches, cell types, and conditions. We further note that we did not apply any feature selection except for filtering out all zero genes. To better guide readers to find this information, we have adjusted the main text under the experiment section to highlight the availability of this information.
>
> > Q7. For the study in Table 2a, is there any consideration for zero-expressed genes?
> >
>
> A. We thank the reviewer for the question. In our original experimental setting, we did not consider zero-expressed genes. However, in the newly added experiments using MCV setting (see our reply to Q1 above), the models are indeed penalized for incorrectly imputing zero-expressed genes to zeros. We showed that under the MCV setting that accounts for zero-expressed genes, as suggested by the reviewer, scDiff outperforms baseline methods (see more details in our reply to Q1).
>
> > Q8. What is the chance level in Figure 2? Is not the number of cell types limited in this context?
> >
>
> A. Thank you for raising your concerns. We have updated the chance level in Figure 2 in the revised manuscript. The number of cell types are limited by the total number of cell types among datasets. Specifically, Liver dataset has in total 13 cell types. We pick the top 5 cell types as majority set (i.e., cell types that have higher number of cells than threshold 600), and the remaining 8 cell types for one-shot evaluation. Thus, we show the results ranging from 2 cell types to 8 cell types as limited by the data.
>
> > Q9. In Figure 3, why the top and bottom subfigures do not reveal the same relative performance pattern? Could you provide further elaboration on how they are related to each other?
> >
>
> A. In Figure 3, the upper panel shows the correlation between the true and predicted gene expression of the perturbation; on the other hand, the bottom panel shows the MSE between the true and predicted (only for the top 20 differentially expressed (DE) genes provided by GEARS). For correlation (upper panel), the higher the better. Conversely, for MSE (bottom panel), the lower the better. We first note that the only performance ranking discrepancy is observed in the Norman benchmark, where scDiff achieved a higher (better) correlation but higher (worse) MSE than GEARS. This discrepancy might occur due to scDiff better capturing the global trend of the perturbed gene expression pattern, while GEARS more accurately captures the exact gene expression value for the top 20 DE genes.

---

> > ### Comment · Reviewer_Z48j · 2023-11-22
> > **Response to the rebuttal**
> >
> > I appreciate the author's efforts in presenting a detailed rebuttal alongside additional experiments, which addressed my questions.
> > However, I would like to revisit my primary concern regarding the paper's novelty and contribution to the field of machine learning. I acknowledge the authors' point that the proposed framework is the first to perform multiple downstream tasks in single-cell analysis by incorporating prior information from ontologies and large language models. However, I believe that the paper's contribution to the broader ML field is relatively limited, given that it primarily focuses on providing a unified supervised framework for single-cell studies. The proposed approach seems like a straightforward conditional extension to the variational diffusion models (Ho et al., 2020, and Kingma et al., 2021).
> >
> > Given the rebuttal effort, I will increase my score to 6.

---

> > > ### Author Response · Authors · 2023-11-22
> > > **Thanks for your response**
> > >
> > > Thanks for your response and support. We are glad to know that our rebuttal has addressed your questions. We acknowledge that our primary contribution is tailored for single-cell analysis rather than the broader ML field. Our aspiration is that our work can serve as a pioneer for applying diffusion models and other advanced ML tools at the forefront of biology and other AI4Science domains. An encouraging next step involves expanding the design space of diffusion models, not only in single-cell analysis but potentially benefiting other fields as well.

---

### Official Review · Reviewer_qLUM · 2023-11-01

**Soundness:** 3 good
**Presentation:** 2 fair
**Contribution:** 3 good
**Rating:** 6
**Confidence:** 3

**Summary:**

- This work applies a conditional diffusion model to scRNA-seq datasets.
 - The proposed model incorporates previously described innovations (e.g. the simplified, unweighted objective, injecting the diffusion time step into embeddings, reverse mixing etc.)
  - Architecture choices are not directly validated/ explored for scRNA-seq datasets.
  - Authors showcase the performance of this model on a range of tasks that fit into the form of estimating $P(X|C)$ where $X$ is expression data, and $C$ is some generic conditioning information.

**Strengths:**

- S1. Authors apply diffusion models to scRNA-seq datasets, and tackle a wide range of tasks.
- S2. At face value the results are quite impressive.

**Weaknesses:**

- W1. The model description is more of a sketch (e.g. even Fig 1) than a self-contained and unambiguous description. More detail about how and what conditioning information is used for each task is lacking.

 - W2. Given that this is an application to a new domain, rather than evaluating it on a suite of tasks with the same mode., it might make sense to characterize the model and the validity of those choices for scRNA-seq datasets.

 - W3. The authors tune their own model, and report performance with default values for competing methods. I appreciate the transparency in communication, but it makes it harder to interpret results. Perhaps using benchmarks where train/test splits are fixed, and other methods are reported with tuned performance would be more informative. [Openproblems](https://openproblems.bio/) curates such benchmarks on a wide variety of such tasks for many easy-to-access datasets. Performing some evaluations with those datasets and comparing the pre-defined metrics on those tasks against the leaderboard there would certainly boost my confidence in results.

 - W4. While there is an impressive number of tests performed with scDiff coming out on top on all fronts, it is unclear why there is a marked improvement. In particular details about the training and testing procedure are missing at a granular-enough level to reproduce results. For example, the Jurkat dataset consists of ~3,000 cells and ~12,000 genes. Was the network trained from scratch on this? Was any preprocessing / feature selection used to report results on that dataset?

 Overall there isn't substantial new methodological development; the performance on scRNAseq datasets is impressive, but manuscript lacks sufficient detail to reproduce them. I'm happy to revise my score if authors are able to satisfactorily address that aspect.

**Questions:**

- Q1. Some of the datasets have a very small number of cells (~3000 for Jurkat). Can the authors clarity in which cases was the model trained from scratch (i.e. with randomly initialized weights)?

 - Q2. Eq. 1 uniform prior assumption seems to be a strong over-simplification. Would this hurt on unbalanced datasets (e.g. when considering cell type classification as in appendix C)?

 - Q3. What is $L$ here? Is it simply the number of distinct conditions?
    > The goal of each conditioner is to extract a set of L numerical representations of an input condition c.

 - Q4. In the same section about the conditioner, can you provide examples for:
    > The mapping here can be designed to suit the specific needs of different input types.

 - Q5. If there are only ~30 cell types in the dataset (e.g. Liver in Fig. 5), how is any high (e.g. 64 or 128 dim) dimensional embedding from LLM's helpful for the model for one-shot classification?

---

> ### Author Response · Authors · 2023-11-17
> **Reply to Reviewer qLUM #1**
>
> > W1. The model description is more of a sketch than a self-contained and unambiguous description. More detail about how and what conditioning information is used for each task is lacking.
>
> A. Thank you for the suggestions. In short, regardless of the task, all types of available conditions in the data are used for training.  Specifically, for a given dataset, we gather all conditions, including cell types, cell batch labels, cell context, and, if available, perturbation status. Along with diffused expression, all the conditions are then fed into *scDiff* to recover the clean expression. We kindly refer the reviewer to Appendix E.1 Table 3 for the available conditions of each dataset.
>
> As detailed in Appendix D, we apply task-specific inference strategies. For annotation, the core idea of the method lies in evaluating the prediction error of the input at various time steps given different cell types. Regarding imputation, we generate samples from the distribution of training data conditioned on the visible values. In perturbation prediction tasks for novel cell types and novel gene perturbations, *scDiff* is trained on the existing conditions in the training set, and the task objective is to generate samples from query conditions.
>
> > W2. Given that this is an application to a new domain, rather than evaluating it on a suite of tasks with the same model, it might make sense to characterize the model and the validity of those choices.
>
> A. To characterize the model and the validity of those choices, we present the ablation studies in annotation task regarding number of layers in embedder and decoder, adding residual connection and removing cross-attention. The results are summarized in the following table, where we report the accuracy on all the annotation datasets across 5 runs. We will elaborate on our motivation for the current model structure according to the results.
>
> The autoencoder structure has shown great success in single-cell analysis [1, 2]. Following the existing works [3, 4], we build our model with a relatively heavy 6-layer encoder and a light 1-layer decoder. Before the encoder, we utilize an embedder to project the input expression into the embedding space. In the encoder, we wish to inject the prior information through different conditions into the expression embeddings, where we choose the cross-attention mechanism due to performance concerns. We extract the embeddings of conditions through various conditioners and feed them into the key and value of the cross-attention blocks. The embedder serves as a projection from expression space to embedding space, while the decoder reverses this process. We empirically observed that a 1-layer neural network is sufficient for such transformation. We note that the model structure has not been optimized for downstream tasks. Tricks like adding U-Net style residual connections can still improve the performance.
> |  |  | Brain | HLCA | Immune | Liver | Pancreas | PBMC12K |
> | --- | --- | --- | --- | --- | --- | --- | --- |
> |  | Default  | 0.9473 +/- 0.0074 | 0.8931 +/- 0.0070 | 0.8442 +/- 0.0076 | 0.8439 +/- 0.0042 | 0.9680 +/- 0.0143 | 0.9670 +/- 0.0042 |
> | Decoder | 2-layer | 0.9445 +/- 0.0041 | 0.8911 +/- 0.0042 | 0.8444 +/- 0.0070 | 0.8589 +/- 0.0238 | 0.9735 +/- 0.0023 | 0.9700 +/- 0.0045 |
> |  | 3-layer | 0.9378 +/- 0.0017 | 0.8994 +/- 0.0038 | 0.8383 +/- 0.0112 | 0.8062 +/- 0.0216 | 0.9731 +/- 0.0014 | 0.9701 +/- 0.0029 |
> |  | 4-layer | 0.9409 +/- 0.0064 | 0.8989 +/- 0.0037 | 0.8414 +/- 0.0051 | 0.8199 +/- 0.0185 | 0.9763 +/- 0.0011 | 0.9744 +/- 0.0002 |
> | Embedder | 2-layer | 0.9394 +/- 0.0029 | 0.8857 +/- 0.0089 | 0.8424 +/- 0.0061 | 0.8434 +/- 0.0239 | 0.9720 +/- 0.0011 | 0.9667 +/- 0.0039 |
> |  | 3-layer | 0.9512 +/- 0.0048 | 0.8847 +/- 0.0053 | 0.8358 +/- 0.0081 | 0.8447 +/- 0.0224 | 0.9745 +/- 0.0026 | 0.9672 +/- 0.0039 |
> |  | 4-layer | 0.9434 +/- 0.0048 | 0.8927 +/- 0.0018 | 0.8461 +/- 0.0014 | 0.8423 +/- 0.0100 | 0.9742 +/- 0.0012 | 0.9606 +/- 0.0036 |
> | General | No cross-attention | 0.9376 +/- 0.0019 | 0.8870 +/- 0.0012 | 0.8318 +/- 0.0028 | 0.8456 +/- 0.0054 | 0.9722 +/- 0.0014 | 0.9608 +/- 0.0042 |
> |  | U-Net residual | 0.9456 +/- 0.0047 | 0.8959 +/- 0.0050 | 0.8502 +/- 0.0075 | 0.8399 +/- 0.0103 | 0.9693 +/- 0.0014 | 0.9644 +/- 0.0021 |
> |  | Remove diffusion | 0.5513 +/- 0.1390 | 0.1487 +/- 0.0313 | 0.1233 +/- 0.1661 | 0.1525 +/- 0.0214 | 0.3749 +/- 0.0627 | 0.5939 +/- 0.1144 |
>
> [1] Lopez, Romain, et al. "Deep generative modeling for single-cell transcriptomics." *Nature methods* 15.12 (2018): 1053-1058.
>
> [2] Eraslan, Gökcen, et al. "Single-cell RNA-seq denoising using a deep count autoencoder." *Nature communications* 10.1 (2019): 390.
>
> [3] Gong, Jing, et al. "xTrimoGene: An Efficient and Scalable Representation Learner for Single-Cell RNA-Seq Data." *NeurIPS* (2023).
>
> [4] He, Kaiming, et al. "Masked autoencoders are scalable vision learners." *Proceedings of the IEEE/CVF conference on computer vision and pattern recognition*. 2022.

---

> ### Author Response · Authors · 2023-11-17
> **Reply to Reviewer qLUM #2**
>
> We also include the ablation study of removing the diffusion process for annotation and imputation in the above table within last reply and the following table, respectively. In this experiment, we retain the model structure in both tasks. Annotation is achieved by selecting the class that minimizes the reconstruction error of the posterior, while we impute the missing values through 1-step prediction from the visible entries. We observe that removing the diffusion process results in a significant performance drop in both tasks, indicating that the diffusion process serves as a crucial part of the proposed probabilistic modeling framework.
> | Corr | Jurkat | 293T | PBMC1K |
> | --- | --- | --- | --- |
> | scDiff | 0.8228 +/- 0.0001 | 0.8110 +/- 0.0002 | 0.7743 +/- 0.0003 |
> | Remove diffusion | 0.7833 +/- 0.0021 | 0.7674 +/- 0.0014 | 0.7439 +/- 0.0004 |
>
> > W3. The authors tune their own model, and report performance with default values for competing methods. [Openproblems](https://openproblems.bio/) curates such benchmarks on a wide variety of such tasks for many easy-to-access datasets. Performing some evaluations with those datasets and comparing the pre-defined metrics on those tasks against the leaderboard there would certainly boost my confidence in results.
>
> A. Thank you for the valuable suggestions. To benchmark *scDiff* with existing leaderboards, we present results follow the setting of OpenProblems in [denoising](https://openproblems.bio/results/denoising/) and [cell type annotation](https://openproblems.bio/results/label_projection/). For annotation task, we copied the accuracy of four top-performing baselines from the [leaderboard](https://openproblems.bio/results/label_projection/) with random splits. The results are as follows, where *scDiff* is evaluated across five runs. We observe that *scDiff* matches the best performance of the leaderboard.
>
> | ACC | Pancreas | Tabula Muris | CeNGEN |
> | --- | --- | --- | --- |
> | Logistic regression (log CP10k) | 0.98 | 0.92 | 0.89 |
> | Seurat reference mapping (SCTransform) | 0.98 | 0.90 | 0.83 |
> | Multilayer perceptron (log scran) | 0.98 | 0.92 | 0.87 |
> | XGBoost (log CP10k) | 0.97 | 0.86 | 0.84 |
> | scDiff | 0.9800 +/- 0.0039 | 0.9257 +/- 0.0066 | 0.8889 +/- 0.0024 |
>
> For denoising task, we selected three performant baselines from the [leaderboard](https://openproblems.bio/results/denoising/) regarding MSE. We report the mean and variance of *scDiff* and baselines across five runs. Note that the MSE results on [leaderboard](https://openproblems.bio/results/denoising/) are scaled for better illustration, we reproduced the benchmark for a fair comparison via the [public code repository](https://github.com/openproblems-bio/openproblems/tree/main/openproblems/tasks/denoising) without scaling. A conclusion can be drawn that *scDiff* outperforms the baselines across datasets.
>
> | MSE | PBMC | Pancreas | Tabula Muris |
> | --- | --- | --- | --- |
> | MAGIC | 0.1887 +/- 0.0001 | 0.2319 +/- 0.0001 | 0.1842 +/- 0.0001 |
> | DCA | 0.2184 +/- 0.0003 | 0.2683 +/- 0.0003 | 0.2160 +/- 0.0007 |
> | ALRA | 0.2823 +/- 0.0007 | 0.3317 +/- 0.0001 | 0.2800 +/- 0.0006 |
> | scDiff | 0.1810 +/- 0.0011 | 0.2079 +/- 0.0027 | 0.1680 +/- 0.0023 |
>
> > W4. While there is an impressive number of tests performed with scDiff coming out on top on all fronts, it is unclear why there is a marked improvement. In particular details about the training and testing procedure are missing at a granular-enough level to reproduce results. For example, the Jurkat dataset consists of ~3,000 cells and ~12,000 genes. Was the network trained from scratch on this? Was any preprocessing / feature selection used to report results on that dataset?
>
> A. Thank you for raising the concern of reproducibility. We formulate our answer in two aspects: (i) data preprocessing, and (ii) model training, inference and implementation.
>
> In the revised manuscript, we add details of data preprocessing in Appendix E.2. For all the datasets, we applied the standard preprocessing pipeline: filtering out all-zero genes and cells, library size normalization with $\text{target sum} = 10,000$ and then logarithm normalization. The dataset statistics can be found in Appendix E.2.
>
> In addition, we elaborate more on model training and inference in Section 2 and Appendix D, and  implementation details in Section 4 and Appendix E. In short, by consolidating all conditions into the prior, we achieve task-agnostic training since the conditions are determined by the datasets rather than the tasks. The types of conditions included for each task are detailed in Appendix E.1 Table 3. As mentioned in Appendix E.1, while keeping the model structure unchanged, we will train a new model from scratch for each task on each dataset except for few-shot annotation setting. Different tasks share a same training strategy but require diverse inference stages. We summarize the task-specific inference strategies in Appendix D.

---

> > ### Comment · Reviewer_qLUM · 2023-11-21
> >
> > Thank you for the detailed response, new benchmarking results, additional clarifications on the model architecture. I have updated my score.

---

> > > ### Author Response · Authors · 2023-11-22
> > > **Thanks for your response**
> > >
> > > Thank you for your response and support. We're pleased to hear that our rebuttal effectively addressed your concerns. Should any further issues arise, please do not hesitate to inform us.

---

> ### Author Response · Authors · 2023-11-17
> **Reply to Reviewer qLUM #3**
>
> > Q1. Some of the datasets have a very small number of cells (~3000 for Jurkat). Can the authors clarity in which cases was the model trained from scratch (i.e. with randomly initialized weights)?
> >
>
> A. We apologize for the obscurity in the manuscript. To clarify, *scDiff* is trained from scratch in each task on each dataset except few-shot cell type annotation. We further refer to the questions posted by reviewer Cfbr, where we concluded that our proposed *scDiff* is well optimized even for small datasets like the three datasets in imputation task. One promising future work is to verify the transferability of *scDiff* pre-trained on large-scale and heterogenous datasets.
>
> > Q2. Eq. 1 uniform prior assumption seems to be a strong over-simplification. Would this hurt on unbalanced datasets (e.g. when considering cell type classification as in appendix C)?
>
> A. We thank the reviewer for the question. We note that the cell type distributions of the current annotation datasets are already quite skewed and unbalanced. In the following table, we calculate the proportion of the number of cells of the largest three and smallest three cell types. The superior cell type annotation performance of *scDiff* highlights the feasibility of the seemingly simplistic uniform prior. Nevertheless, we agree that more careful handling to take into account of the non-uniformity could be an interesting avenue to explore in the future.
>
> | Proportion | PBMC12K | Pancreas | HLCA | Immune | Brain | Liver |
> | --- | --- | --- | --- | --- | --- | --- |
> | Majority 1 | 0.4167 | 0.3353 | 0.1445 | 0.1125 | 0.4081 | 0.4143 |
> | Majority 2 | 0.1857 | 0.2545 | 0.1348 | 0.0899 | 0.1047 | 0.1320 |
> | Majority 3 | 0.1352 | 0.1308 | 0.1061 | 0.0801 | 0.0810 | 0.0973 |
> | Minority 3 | 0.0293 | 0.0020 | 0.0005 | 0.0015 | 0.0058 | 0.0143 |
> | Minority 2 | 0.0283 | 0.0015 | 0.0004 | 0.0011 | 0.0035 | 0.0063 |
> | Minority 1 | 0.0073 | 0.0004 | 0.0003 | 0.0010 | 0.0023 | 0.0024 |
>
> > Q3. What is L here? Is it simply the number of distinct conditions?
> >
> >
> > > The goal of each conditioner is to extract a set of L numerical representations of an input condition c.
> > >
>
> A. We thank the reviewer for pointing out the vague definition of the number of conditions $L$. We confirm that $L$ is indeed the number of conditions, and we have updated the manuscript to clarify this further.
>
> > Q4. In the same section about the conditioner, can you provide examples for:
> >
> >
> > > The mapping here can be designed to suit the specific needs of different input types.
> > >
>
> A. Thank you for the suggestion regarding the examples of condition mappings. Here we provide some examples. The matrix-valued observed counts in imputation will go through a linear transformation; The class labels of cells are represented by learnable embeddings; The cell ontology textual descriptions will be processed by LLMs; The gene ontology graph will be fed into graph neural networks. We have added a few examples following the aforementioned sentence in the revised manuscript.
>
> > Q5. If there are only ~30 cell types in the dataset (e.g. Liver in Fig. 5), how is any high (e.g. 64 or 128 dim) dimensional embedding from LLM's helpful for the model for one-shot classification?
>
> A. We thank the reviewer for the question regarding why LLM helps in the few-shot learning setting. In short, LLM helps by providing prior knowledge about the relationships between different cell types, potentially bridging the training and few-shot testing cell types. BioLinkBERT (the LLM we used in our paper) is trained on a vast biomedical corpus, which contains many medical terms (such as cell types) and descriptions of how different terms relate. The trained BioLinkBERT model can thus provide inductive bias that reflects how similar two cell types are given their biomedical definitions (extracted from the cell ontology). We argue that this inductive bias is the key to its success in enhancing the few-shot annotation setting. For example, during training, the model may have access to training examples of B-cells. Suppose during the few-shot training, the two target cell types are plasma cells (a specific type of B-cell) and neuron cells (utterly distinct from the B-cell lineage). Then, BioLinkBERT can generate the cell type embedding of plasma cells that are more similar to the B-cell but less similar to the Neuron cell. This distinction is helpful for the model to classify the two different cell types correctly.

---

### Official Review · Reviewer_Cfbr · 2023-11-03

**Soundness:** 4 excellent
**Presentation:** 4 excellent
**Contribution:** 4 excellent
**Rating:** 8
**Confidence:** 5

**Summary:**

The authors present scDiff, a diffusion model for single cell sequencing tasks.
They start off by phrasing a variety of tasks within the single cell sequencing world as probabilistic modeling tasks, which may involve a shared generative model of single cells and a generic mechanism for conditioning.
Once they establish that -correct- link, they proceed to utilize diffusion models to build such a generic "prior" over single cells, and pair it with a conditioning mechanism per layer in order to be able to inject specific conditions or knowledge into the model.
Once they establish that modeling framework, which closely follows the diffusion modeling best practices, the authors proceed to test their model in a variety of tasks by utilizing conditioning in various was.

In their tasks ranging from conditioning on prior knowledge, cell type inference, and imputation, they show that the model is broadly applicable and performs well.

**Strengths:**

I want to praise this paper, it does a lot of things well that I've been hoping to see in the field.

Casting single cell sequencing modeling as a generative model is not a novelty per se, various VAE frameworks have blazed that trail, but utilizing a diffusion model sheds a particular light on this that is both useful and performant.

This paper stands well as a baseline introduction into what can be done with diffusion models as they currently are and executes neatly on the idea of using the conditioning model per layer to map to a variety of tasks.
It is also important that the paper insists on the unified model ov er all these tasks and realizes that conditioning is the one mechanism to express different tasks, but unifies the model which can modularly be decomposed from the conditioning aspect.

The paper is also overall well written and attributes ideas well, and the application domain will benefit from its existence.

**Weaknesses:**

I have three nitpicks to note in the paper.

First, the application here is straightforward, no core ML innovation was necessary to execute this project.
However, I think that realizing how diffusion models map to this important modeling domain and executing the basics well once absolutely justifies this well-executed paper, but I wanted to note that it is not innovating dramatically.

Second, the authors keep using the term "casting single cell tasks as posterior inference" throughout the paper, but do not really perform much inference in truth beyond the classical training scenario of diffusion models.
I would prefer if they used language like "we cast these tasks as probabilistic modeling using a shared model", since inference is not the heart of the story here and in fact is relatively generically solved since we do not really inspect posterior distributions per layer over specific conditions and so on.  This is a minor point, but since I anticipate this paper to be read a lot it would be good to use that language carefully.

If I can add a third nitpick, the imputation task is not entirely correct in how it is phrased as a conditioning setting.
If the authors inspect their predictive distribution, it probably contains outputs beyond the ones that are visible.
I understand the conditioning mechanism injected per layer will increase affinity towards imputing the right thing, but it's not exactly p(x|m*x) that the authors are modeling, but rather a p(x*|m*x) where x* is sampled from "some" distribution conditioned on embeddings of x*m.
I want to point out that I find this to be a fine approximation to the task and not a reason to reject the paper, but I would prefer the authors to call it out as such and leave space for future improvements.

**Questions:**

I would be curious to see the scaling behavior of the systems the authors study given different dataset sizes.

Diffusion models tend to be data and compute hungry, how do they behave here? Can we apply them on one screen?
Do we have to pre-retrain them broadly?

I am sold on the modeling framework and think this paper will stand on its own, but the field would get more value out of it if we inspected these questions of data efficiency.

---

> ### Author Response · Authors · 2023-11-17
> **Reply to Reviewer 1 Cfbr #1**
>
> We sincerely appreciate your recognition of this work. In the following content, we will try to address your comments and suggestions.
>
> > W1. The application here is straightforward, no core ML innovation was necessary to execute this project. However, I think that realizing how diffusion models map to this important modeling domain and executing the basics well once absolutely justifies this well-executed paper, but I wanted to note that it is not innovating dramatically.
>
> A. We appreciate the reviewer's recognition of our efforts in adapting diffusion models to single-cell analysis. While we concur that our work may not lie in innovating core ML, we emphasize its novelty in applying these models to a new domain. Our approach not only introduces a fresh perspective to single-cell analysis but also demonstrates the untapped potential of diffusion models in this field.
> Our work serves as a bridge, connecting advanced diffusion model research with pressing challenges in single-cell analysis. We believe this interdisciplinary fusion is an innovation in its own right, offering new methodologies and insights that enrich both fields. The practical implications and potential for future research stemming from our unified framework underscore its significance as a meaningful contribution to the scientific community.
>
> > W2. The authors keep using the term "casting single cell tasks as posterior inference" throughout the paper, but do not really perform much inference in truth beyond the classical training scenario of diffusion models. I would prefer if they used language like "we cast these tasks as probabilistic modeling using a shared model", since inference is not the heart of the story here and in fact is relatively generically solved since we do not really inspect posterior distributions per layer over specific conditions and so on. This is a minor point, but since I anticipate this paper to be read a lot it would be good to use that language carefully.
>
> A. We thank the reviewer for the insightful feedback regarding our terminology. In this work, we refer to our framework as *posterior estimation* to highlight that we are modeling the *posterior* of gene expression. This emphasizes the discrepancy between the proposed framework and the deterministic formulation of the downstream tasks. We agree that the inference stage is not the core of the story and the exact likelihood computation is not necessary regarding the downstream tasks. To be consistent, we have updated the manuscript by replacing *posterior estimation* with *posterior modeling* and *probabilistic modeling*.
>
> > W3. The imputation task is not entirely correct in how it is phrased as a conditioning setting. If the authors inspect their predictive distribution, it probably contains outputs beyond the ones that are visible. I understand the conditioning mechanism injected per layer will increase affinity towards imputing the right thing, but it's not exactly p(x|m*x) that the authors are modeling, but rather a p(x*|m*x) where x* is sampled from "some" distribution conditioned on embeddings of x*m. I want to point out that I find this to be a fine approximation to the task and not a reason to reject the paper, but I would prefer the authors to call it out as such and leave space for future improvements.
>
> A. Thank you for carefully inspecting our formulation of the downstream tasks. We agree that there is a gap between the generated samples and the missing values. We incorporate cell context as a condition during the training stage to model the posterior of the training data. In the generation stage, we approximate the missing values by sampling from the distribution of training data conditioned on the visible values. We add a footnote in the updated manuscript for clarification.

---

> > ### Author Response · Authors · 2023-11-17
> > **Reply to Reviewer 1 Cfbr #2**
> >
> > > Q1. I would be curious to see the scaling behavior of the systems the authors study given different dataset sizes.
> >
> > A. Thank you for the valuable question about scaling of dataset size. We conducted a preliminary experiment to examine whether *scDiff* follows the scaling law regarding the number of cells in training data. Starting from the three large datasets we used in annotation task, we subsampled the training sets by randomly keeping a proportion of training cells and evaluate *scDiff* on the same testing sets. We summarize the annotation accuracy across five runs in the following table, where the subsample ratios range from 0.01 to 1.0.
> >
> > | Subsample ratio | HLCA | Immune | Brain |
> > | --- | --- | --- | --- |
> > | 0.01 | 0.2839 +/- 0.0135 | 0.3275 +/- 0.0238 | 0.7797 +/- 0.0035 |
> > | 0.05 | 0.5924 +/- 0.0077 | 0.5679 +/- 0.0239 | 0.7673 +/- 0.0048 |
> > | 0.1 | 0.8253 +/- 0.0189 | 0.6995 +/- 0.0128 | 0.8951 +/- 0.0403 |
> > | 0.2 | 0.8916 +/- 0.0022 | 0.8020 +/- 0.0096 | 0.9510 +/- 0.0056 |
> > | 0.3 | 0.8951 +/- 0.0064 | 0.8362 +/- 0.0036 | 0.9659 +/- 0.0044 |
> > | 0.4 | 0.8807 +/- 0.0079 | 0.8598 +/- 0.0013 | 0.9599 +/- 0.0019 |
> > | 0.5 | 0.8799 +/- 0.0059 | 0.8625 +/- 0.0069 | 0.9279 +/- 0.0058 |
> > | Full | 0.8931 +/- 0.0070 | 0.8442 +/- 0.0076 | 0.9473 +/- 0.0074 |
> >
> > For all three datasets, we observe that the accuracy has already matched or even outperformed that of full dataset before the subsample ratio reaches 0.5. A possible explanation is that the scRNA-seq data is not as diverse as images in vision. This suggests that *scDiff* trained from scratch does not require large amount of training dataset for the annotation task. A possible next step is to check how cross-dataset pre-training will affect the results.
> >
> > > Q2. Diffusion models tend to be data and compute hungry, how do they behave here? Can we apply them on one screen? Do we have to pre-retrain them broadly?
> >
> > A. As depicted in the above table, *scDiff* achieves comparable or even higher accuracy in annotation before the subsample ratio reaches 0.5. In addition, it is worth noting that our *scDiff* is trained from scratch in each task on each dataset. It outperforms the baselines even in the small datasets (e.g., three imputation datasets have at most ~3000 cells). We can have a preliminary conclusion that diffusion model is not as data hungry as they are in vision domain.
> >
> > We note that *scDiff* has a computational complexity that is nearly the same as a standard MLP. In the following table, we report the per-epoch computation time for scDiff against standard baselines for all our benchmarking tasks.  This supports that *scDiff* can be scaled to large datasets for possible pre-training. The experiment in Q1 indicates that increasing the dataset size has limited improvement for annotation task. However, it is still unclear whether cross-dataset pre-training will help to boost the performance in annotation and other downstream tasks. We leave it as a future work to explore the possibilities of large-scale pre-training for *scDiff.*
> >
> > |  | Dataset | scDiff per-epoch runtime (sec) | Baseline per-epoch runtime (sec) | Baseline name |
> > | --- | --- | --- | --- | --- |
> > | Annotation | Brain | 9.63 | 9.82 | scANVI |
> > |  | HLCA | 13.60 | 12.73 | scANVI |
> > |  | Immune | 5.28 | 8.86 | scANVI |
> > |  | Liver | 2.83 | 2.97 | scANVI |
> > |  | Pancreas | 5.05 | 14.37 | scANVI |
> > |  | PBMC12K | 2.04 | 3.31 | scANVI |
> > | Denoising | 293T | 0.8847 | 0.53 | DCA |
> > |  | Jurkat | 0.5736 | 0.60 | DCA |
> > |  | PBMC1K | 0.2634 | 0.16 | DCA |
> > | Perturbation | HPoly | 1.96 | 1.83 | scGEN |
> > |  | PBMC | 3.16 | 3.53 | scGEN |
> > |  | Salmonella | 1.58 | 1.81 | scGEN |
> > | Gene Perturbation | Adamson | 83.88 | 1189.66 | GEARS |
> > |  | Dixit | 44.67 | 760.70 | GEARS |
> > |  | Norman | 155.88 | 491.97 | GEARS |

---

### Author Response · Authors · 2023-11-20
**Friendly Request for Discussion**

Dear Reviewers,

We wish to express our gratitude for your thoughtful comments and concerns. Following your valuable feedback, we have submitted our responses and revisions to address the issues. As the discussion period is about to end, we kindly request your confirmation of the receipt of our responses. Additionally, we welcome any further concerns or suggestions regarding our revision. Your timely response is greatly appreciated and will be immensely helpful for us to improve our work.


Thank you for your time and consideration.


Sincerely,

The Authors

---

### Meta-Review · Area_Chair_bWqy · 2023-12-05

**Metareview:**

The paper presents an approach for single-cell analysis using diffusion models. The proposed approach scDiff can solve prediction tasks, imputation tasks, and general transfer learning problems.

Although the paper general received borderline positive to positive reviews, one concern is that the paper sort of lies in the middle -- it lacks novelty on the ML side but also lacks a rigorous qualitative evaluation (pretty much all of the experiments are on quantitative/predictive tasks).

On the topic of single-cell analysis, there are prior works like scVI, scVAE, scDREAMER, totalVI, etc that have appeared in venues such as nature biotech or nature communication. These papers also don't have a lot of novelty on the ML side (mostly take existing deep generative models and adapt them for single-cell analysis tasks) but the experimental evaluations in these papers have much more depth.

scVI: https://www.nature.com/articles/s41592-018-0229-2
scVAE: https://academic.oup.com/bioinformatics/article/36/16/4415/5838187
scDREAMER: https://www.nature.com/articles/s41467-023-43590-8
totalVI: https://www.nature.com/articles/s41592-020-01050-x

The paper does claim about the "generality" aspect (that it can solve several type of problems such as prediction, imputation, transfer learning), but that in itself is not a big aspect because that is usually possible anyway with generative models.

My concern is that, regardless of whether the paper is submitted to an ML venue or a more bio-centric venue, at least one of the aspects (ML part or the evaluation part) should be solid, which is not quite the case with this paper.

**Justification For Why Not Higher Score:**

The paper lacks novelty in terms of the ML methodology and the empirical analysis is also weak (mostly focused on quantitative results).

**Justification For Why Not Lower Score:**

N/A

---

### Decision · Program_Chairs · 2024-01-16

Reject